# Chromosomal instability in aneuploid acute lymphoblastic leukemia associates with disease progression

Oscar Molina [1,2,✉], Carmen Ortega-Sabater[3,21], Namitha Thampi [1,2,21], Narcís Fernández-Fuentes [1,2], Mercedes Guerrero-Murillo [1,2], Alba Martínez-Moreno [1,2], Meritxell Vinyoles [1,2], Talía Velasco-Hernández [1,2], Clara Bueno[1,2], Juan L Trincado [1,2], Isabel Granada[4,5], Diana Campos[6], Carles Giménez[6], Judith M Boer [7], Monique L den Boer[7,8], Gabriel F Calvo [3], Mireia Camós[9,10,11], Jose-Luis Fuster [12], Pablo Velasco[13], Paola Ballerini [14], Franco Locatelli [15], Charles G Mullighan [16], Diana CJ Spierings [17], Floris Foijer [17], Víctor M Pérez-García [3] & Pablo Menéndez [1,2,18,19,20✉]

## Abstract

Chromosomal instability (CIN) lies at the core of cancer development leading to aneuploidy, chromosomal copy-number heterogeneity (chr-CNH) and ultimately, unfavorable clinical outcomes. Despite its ubiquity in cancer, the presence of CIN in childhood B-cell acute lymphoblastic leukemia (cB-ALL), the most frequent pediatric cancer showing high frequencies of aneuploidy, remains unknown. Here, we elucidate the presence of CIN in aneuploid cB-ALL subtypes using single-cell whole-genome sequencing of primary cB-ALL samples and by generating and functionally characterizing patient-derived xenograft models (cB-ALL-PDX). We report higher rates of CIN across aneuploid than in euploid cB-ALL that strongly correlate with intraclonal chr-CNH and overall survival in mice. This association was further supported by in silico mathematical modeling. Moreover, mass-spectrometry analyses of cB-ALL-PDX revealed a "CIN signature" enriched in mitotic-spindle regulatory pathways, which was confirmed by RNA-sequencing of a large cohort of cB-ALL samples. The link between the presence of CIN in aneuploid cB-ALL and disease progression opens new possibilities for patient stratification and offers a promising new avenue as a therapeutic target in cB-ALL treatment.

**Keywords** Chromosomal Instability; Aneuploidy; Childhood B-cell Acute Lymphoblastic Leukemia; Disease Models

**Subject Categories** Cancer; Chromatin, Transcription & Genomics; Haematology

## Introduction

Chromosomal instability (CIN), defined as an increased rate of chromosome segregation errors during cell division, is a prominent form of genomic instability (Bakhoum and Landau, 2017) and a major cause of aneuploidy, the most prevalent genetic alteration in human cancers (Ben-David and Amon, 2020; Vasudevan et al, 2021). Aneuploidy is commonly associated with ongoing CIN through consecutive cell divisions (Garribba et al, 2023; Santaguida and Amon, 2015), resulting in intratumor genetic heterogeneity, a central driver of cancer evolution and therapeutic resistance (Ben-David and Amon, 2020; Sansregret et al, 2018). The ability to cope with the ongoing genetic imbalances caused by CIN is a fundamental difference between malignant and non-malignant cells. These imbalances result in several aneuploidy-associated stressors that impair overall cellular fitness such as metabolic changes affecting the protein turnover machinery, replication stress

[1]Josep Carreras Leukemia Research Institute, Department of Biomedicine, School of Medicine, University of Barcelona, Barcelona, Spain. [2]Red Española de Terápias Avanzadas (TERAV), Instituto de Salud Carlos III, Barcelona, Spain. [3]Mathematical Oncology Laboratory, Department of Mathematics & Institute of Applied Mathematics in Science and Engineering, Universidad de Castilla-La Mancha, Ciudad Real, Spain. [4]Hematology Service, Institut Català d'Oncologia (ICO)-Hospital Germans Trias i Pujol, Badalona, Spain. [5]Josep Carreras Leukemia Research Institute, Autonomous University of Barcelona, Badalona, Spain. [6]DiNA Science, S.A, Ciudad Real, Spain. [7]Princess Maxima Center for Pediatric Oncology, Utrecht, The Netherlands. [8]Department of Pediatric Oncology and Hematology, Erasmus Medical Center – Sophia Children's Hospital, Rotterdam, The Netherlands. [9]Hematology Laboratory, Hospital Sant Joan de Déu, University of Barcelona, Barcelona, Spain. [10]Leukemia and Other Pediatric Hemopathies, Developmental Tumor Biology Group, Institut de Recerca Hospital Sant Joan de Déu, Barcelona, Spain. [11]Centro de Investigación Biomédica en Red de Enfermedades Raras (CIBERER), Instituto de Salud Carlos III, Madrid, Spain. [12]Pediatric Hematology and Oncology Department, Hospital Clínico Universitario Virgen de la Arrixaca, Instituto Murciano de Investigación Biosanitaria (IMIB), Murcia, Spain. [13]Pediatric Oncology and Hematology Department, Hospital Vall d'Hebrón, Barcelona, Spain. [14]AP-HP, Service of Pediatric Hematology, Hopital Armand Trousseau, Paris, France. [15]Bambino Gesù Children's Hospital, Catholic University of Sacred Heart, Rome, Italy. [16]Department of Pathology, St. Jude Children's Research Hospital, Memphis, TN, USA. [17]European Research Institute for the Biology of Aging (ERIBA), University of Groningen, University Medical Center Groningen, Groningen, The Netherlands. [18]Institució Catalana de Recerca i Estudis Avançats (ICREA), Barcelona, Spain. [19]Department of Biomedicine. School of Medicine, University of Barcelona, Barcelona, Spain. [20]Spanish Cancer Research Network (CIBERONC), ISCIII, Barcelona, Spain. [21]These authors contributed equally: Carmen Ortega-Sabater, Namitha Thampi. ✉E-mail: omolina@carrerasresearch.org; pmenendez@carrerasresearch.org

affecting genome integrity, and alterations in the mitotic regulatory machinery which fuel chromosome mis-segregation (Cohen-Sharir et al, 2021; Crasta et al, 2012, Donnelly et al, 2014; Garribba et al, 2023; Ohashi et al, 2015; Oromendia and Amon, 2014; Stingele et al, 2012; Torres et al, 2007). This suggests that CIN must be maintained under a tolerable threshold to preserve cancer cell homeostasis, which may represent a key vulnerability of aneuploid tumors. Indeed, strategies to modulate CIN levels in cancer cells have been therapeutically explored (McClelland, 2017). For instance, increasing CIN levels in tumors with ongoing CIN has been explored as a strategy to drive them over the threshold of tolerance, and an increased efficacy of some cancer treatments (such as paclitaxel and radiation therapy) that enhance CIN in cells with high basal levels of CIN has been reported (Bakhoum et al, 2015; Janssen et al, 2009). Inversely, decreasing CIN levels by manipulating key mitotic pathways such as the spindle-assembly checkpoint (SAC) and microtubule dynamics resulted in a significant improvement in overall survival (OS) in preclinical mouse models (Bakhoum et al, 2009; Cohen-Sharir et al, 2021; Ertych et al, 2014; Orr et al, 2016; Sansregret et al, 2017). In fact, phase I clinical studies assessing the efficiency of inhibitors of MPS1, a master regulator of the SAC, and of KIF18A, a kinesin-like motor protein that regulates chromosome positioning during cell division, are currently being conducted to treat a variety of cancer types (NCT02366949, NCT04293094).

Despite the ubiquitous presence of CIN in several aneuploid cancer types and its clinical relevance, its presence in B-cell acute lymphoblastic leukemia (B-ALL) remains largely unexplored. B-ALL is the most frequent childhood cancer and it is characterized by the accumulation of highly proliferative immature B-cell precursors in the bone marrow (BM) (Hunger and Mulligan, 2015). B-ALL is a genetically heterogeneous disease with distinct biological and prognostic subgroups classified according to cytogenetic and molecular features (Brady et al, 2022; Moorman, 2012). Among them, aneuploidy is the most common genetic abnormality in B-ALL, particularly in childhood B-ALL (cB-ALL), with ~35–40% of cases showing abnormal chromosome numbers in leukemic cells, which is considered as an important prognostic factor (Molina et al, 2021a). Clinically relevant aneuploid subtypes include high hyperdiploidy (HeH), the most frequent cB-ALL subtype, defined by the presence of 51 to 67 chromosomes in leukemic cells and associated with a favorable outcome (5-year overall OS > 85%) (Haas and Borkhardt, 2022). Low hypodiploidy (HoL), with 30 to 39 chromosomes, and near-haploidy (NH), with 24 to 29 chromosomes, account for ~2% of cB-ALL cases and have an extremely poor prognosis, with a 5-year OS of <20% (Molina et al, 2021b). Despite being very uncommon in cB-ALL, most HoL patients harbor inherited *TP53* mutations, suggesting an evolution of a Li-Fraumeni syndrome (Holmfeldt et al, 2013). Noteworthy, half of all the HoL and NH cases ultimately show a chromosomal doubling of the initial hypodiploid clone, resulting in hyperdiploid clones with 50 to 78 chromosomes that frequently represent the major leukemic clone at diagnosis (Harrison et al, 2004; Holmfeldt et al, 2013). This poses an important clinical challenge, as these patients could be erroneously classified and treated as HeH-B-ALL despite being at higher risk of treatment failure (Nachman et al, 2007). Chromosomal gains and losses in aneuploid cB-ALL are not random, with specific chromosomes preferentially gained in HeH or retained as disomies

in HoL and NH-B-ALL (Molina et al, 2021a), suggesting a potential leukemogenic impact of these chromosomes.

The presence of CIN and its contribution to aneuploid cB-ALL progression is largely unknown due to the lack of preclinical models to study actively dividing cells. Accordingly, studies of CIN in cB-ALL are limited to the characterization of chromosomal copy-number heterogeneity (chr-CNH) in primary cB-ALL samples and remain controversial due to the different techniques used to assess karyotype variability (Alpar et al, 2014; Heerema et al, 2007; Paulsson et al, 2010; Ramos-Muntada et al, 2022; Talamo et al, 2010). Importantly, although increased rates of chromosome mis-segregation have been shown in actively dividing HeH-B-ALL cells in patient-derived xenograft (PDX) models (Molina et al, 2020), they are vastly unexplored in other clinically relevant aneuploid subtypes, such as HoL- and NH-B-ALL. These methods provide insight into the genomic complexity of cancer genomes but do not allow for an assessment of whether CIN is ongoing, or whether errors are tolerated and/or efficiently propagated. Thus, it remains unresolved whether aneuploid cB-ALL subtypes experience ongoing CIN, whether the extent of CIN correlates with karyotype heterogeneity, and whether CIN influences leukemia progression.

Here, we explored the presence and the levels of CIN in different clinically relevant aneuploid subtypes of cB-ALL using single-cell whole-genome sequencing (WGS) of primary samples to reliably assess chr-CNH, and by systematically generating PDX models from primary cB-ALL samples (cB-ALL-PDX). These models allowed us to integrate chromosomal segregation data of actively dividing cells in the BM with karyotype heterogeneity and disease progression. Overall, our results reveal variable levels of CIN in aneuploid cB-ALL subtypes, which significantly correlate with intraclonal karyotype heterogeneity and with disease progression. In addition, mass-spectrometry analyses of cB-ALL-PDX samples revealed a "CIN signature" enriched in mitosis and chromosome segregation regulatory pathways. We speculate that this signature identifies adaptive mechanisms to ongoing CIN in aneuploid cB-ALL cells, which displayed a transcriptional signature characterized by an impaired mitotic spindle as observed by RNA-sequencing (RNA-Seq) analyses of a large cohort of primary cB-ALL patient samples. Our work contributes to improve stratification of patients with cB-ALL with different levels of CIN who could benefit from new therapeutic approaches aiming to target ongoing CIN.

## Results

### Chr-CNH is consistently higher in aneuploid subtypes of childhood B-ALL

Chr-CNH, defined as cell-to-cell variability in whole chromosomes and chromosome arms, is a major readout of CIN (Bakhoum and Landau, 2017). Although chr-CNH is usually associated with aneuploidy, its presence in the different aneuploid subtypes of cB-ALL is debated due to technical limitations commonly used to assess karyotype variability (Alpar et al, 2014; Elghezal et al, 2001; Heerema et al, 2007; Paulsson et al, 2010; Ramos-Muntada et al, 2022; Talamo et al, 2010). To comprehensively assess chr-CNH in aneuploid cB-ALL patients, we first applied a recently described computational approach to infer chr-CNH from bulk WGS data

which relies on the deviation from strictly integer chromosomal numbers in WGS samples to calculate chr-CNH (van Dijk et al, 2021) (Fig. 1A). We collected WGS data from the Pediatric Cancer Genome Project (SJC-DS-1001; https://platform.stjude.cloud/data/cohorts) and applied this approach on the patient data. Patients were classified as euploid or aneuploid (HeH, HoL, and NH) subtypes based on available cytogenetic data and/or copy-number alterations identified by RNA-Seq (Dataset EV1). Results showed that chr-CNH was moderately but significantly higher in aneuploid cB-ALL patient samples than in equivalent samples with euploid karyotypes ($P < 0.01$) (Fig. 1B).

To unambiguously identify and characterize karyotype heterogeneity in aneuploid cB-ALL, we next performed low-pass single-cell WGS (scWGS) in 8 primary cB-ALL samples, including two samples for each ploidy group (Eup-, HeH-, HoL-, and NH-B-ALL) (Fig. 1C and Table 1). We also included a pool of healthy human hematopoietic stem/progenitor cells (HSPCs) as a chromosomally stable control (Fig. 1D). Results of scWGS showed remarkably consistent data with the karyotypes obtained in primary B-ALL samples (Fig. 1C and Appendix Table S1). Diploid cells were observed in 1 of 15 cells from patient HeH1 and 5 of 16 cells from patient HoL2, most likely representing healthy hematopoietic cells. Remarkably, both hypodiploid and doubled-up hyperdiploid clones could be observed in patient NH1, and masked hypodiploidy with an exact duplication of the original hypodiploid clone detected by FISH could be observed in patient NH3 (Fig. 1C and Table 1). Chr-CNH, as detected by a genome-wide heterogeneity score (HS), was similarly low in both Eup-B-ALL samples and HSPCs (HS = 0.081) (Fig. 1C,D), indicating that euploid cB-ALL samples are chromosomally stable. Contrastingly, whole-chromosome gains and losses deviating from the modal karyotypes could be observed in all aneuploid cB-ALL samples, with higher genome-wide HS (ranging from 0.141 to 0.266) when compared with Eup-B-ALL and HSPCs. Of note, hypodiploid subtypes (HoL- and NH-B-ALL) showed higher levels of chr-CNH than HeH-B-ALL samples (Fig. 1C). The HS observed across aneuploid cB-ALL samples are well within the range of those observed in previous studies using scWGS of cB-ALL samples (Bakker et al, 2016; Woodward et al, 2023), demonstrating the reliability of our data. Overall, the results consistently showed moderately higher levels of chr-CNH in all aneuploid cB-ALL samples than in Eup-B-ALL and HSPCs. The different levels of CIN across the different aneuploid cB-ALL subtypes might underlie their different clinical outcomes.

## Patient-derived xenograft (PDX) models recapitulate the clinical outcome of cB-ALL in patients with variable ploidy levels

CIN is caused by an increased frequency of chromosome segregation errors, leading to cell-to-cell variability in chromosomal content and in adaptation to diverse cellular stresses (Vasudevan et al, 2021). Given the inherent complexity in studying the actual rate of chromosome mis-segregation in actively dividing cB-ALL primary cells, CIN is typically assessed by quantifying chr-CNH within a given population (Alpar et al, 2014; Ramos-Muntada et al, 2022; Woodward et al, 2023). To comprehensively study the presence of CIN and its relationship with both chr-CNH and disease outcome, we generated PDX models using a discovery cohort consisting of 12 primary diagnostic cB-ALL samples, three

samples per ploidy group (Table 1 and Fig. 2A). A total of 56 cB-ALL-PDX models were generated, with a minimum of three PDX for each primary cB-ALL sample. The follow-up of human engraftment in the peripheral blood (PB) of mice revealed differences in the leukemia kinetics across cB-ALL subgroups, as observed by the rates of early- and late-engrafting PDX, defined as mice with >15% or <15% human leukemic cells in PB at week 12, respectively (Fig. 2B). The frequency of early-engrafting PDX was higher in the HoL- and NH-B-ALL groups (84.6% and 64.3%, respectively) than in the HeH-B-ALL and Eup-B-ALL groups (43.8% and 15.4%, respectively). Remarkably, the HeH-B-ALL group was the most heterogeneous in terms of leukemia kinetics (Fig. 2B). We next correlated leukemia kinetics with clinical outcomes. Results revealed significant differences between HoL- and NH-B-ALL as compared with HeH- and Eup-B-ALL groups for both event-free survival (EFS) and OS rates (Fig. 2C,D), recapitulating the less favorable clinical outcomes of these subtypes of cB-ALL (Haas and Borkhardt, 2022; Molina et al, 2021b). Of note, no differences were observed in the ratios of proliferating and apoptotic B-ALL blasts between PDX groups (Appendix Fig. S1), indicating that other factors (such as CIN) might be involved in the different leukemia kinetics and clinical outcomes observed. Overall, cB-ALL-PDX mirror the clinical outcomes of patients with cB-ALL.

## Aneuploid cB-ALL subtypes are characterized by higher rates of mitotic defects and chromosome mis-segregation

To examine chromosome segregation fidelity in primary cB-ALL cells, we made use of 36 PDX-B-ALL models generated from 12 cB-ALL patients (three individual PDXs per primary cB-ALL sample) (Table 1). Xenografted cB-ALL blasts were isolated and processed for immunofluorescence staining for DNA, kinetochores, and mitotic spindles (Molina et al, 2020). Analyses revealed an increased mitotic index in all aneuploid cB-ALL subtypes when compared with Eup-B-ALL, as observed by the rates of mitotic cells in a minimum of 1500 blasts per sample (mean mitotic index±SEM: >1.5% ± 0.2 vs 0.8% ± 0.06, respectively; $P < 0.01$) (Fig. 3A). Despite higher mitotic indices, only marginal differences were observed in the mitotic progression among cB-ALL groups, as observed by the rates of cells in specific mitotic phases (Fig. 3B), suggesting higher proliferation rates in aneuploid cB-ALL blasts than in Eup-B-ALL blasts.

To assess the levels of CIN, we next analyzed the rates of mitotic and chromosome segregation defects. Different mitotic defects were observed, including chromosome bridges, lagging chromosomes, mitotic spindle defects, misaligned chromosomes at the metaphase plate, and other defects mainly involving unequal cytokinesis (Fig. 3C). Mean rates of mitotic defects in Eup-B-ALL PDX were low (1.4% ± 0.25) while higher and variable rates were observed across aneuploid cB-ALL subtypes, ranging from 3.5% to 17% (Figs. 3D and EV1A,B). Of note, although variable rates of specific mitotic defects were observed throughout cB-ALL subgroups, chromosome alignment defects in the metaphase plates and multipolar spindles (detected as more than two pericentrin foci) were the most frequent abnormalities in aneuploid cB-ALL cells (Figs. 3D and EV1B), in line with our previous observations in an independent cohort of HeH-B-ALL PDX samples (Molina et al, 2020). Unequal cytokineses were more

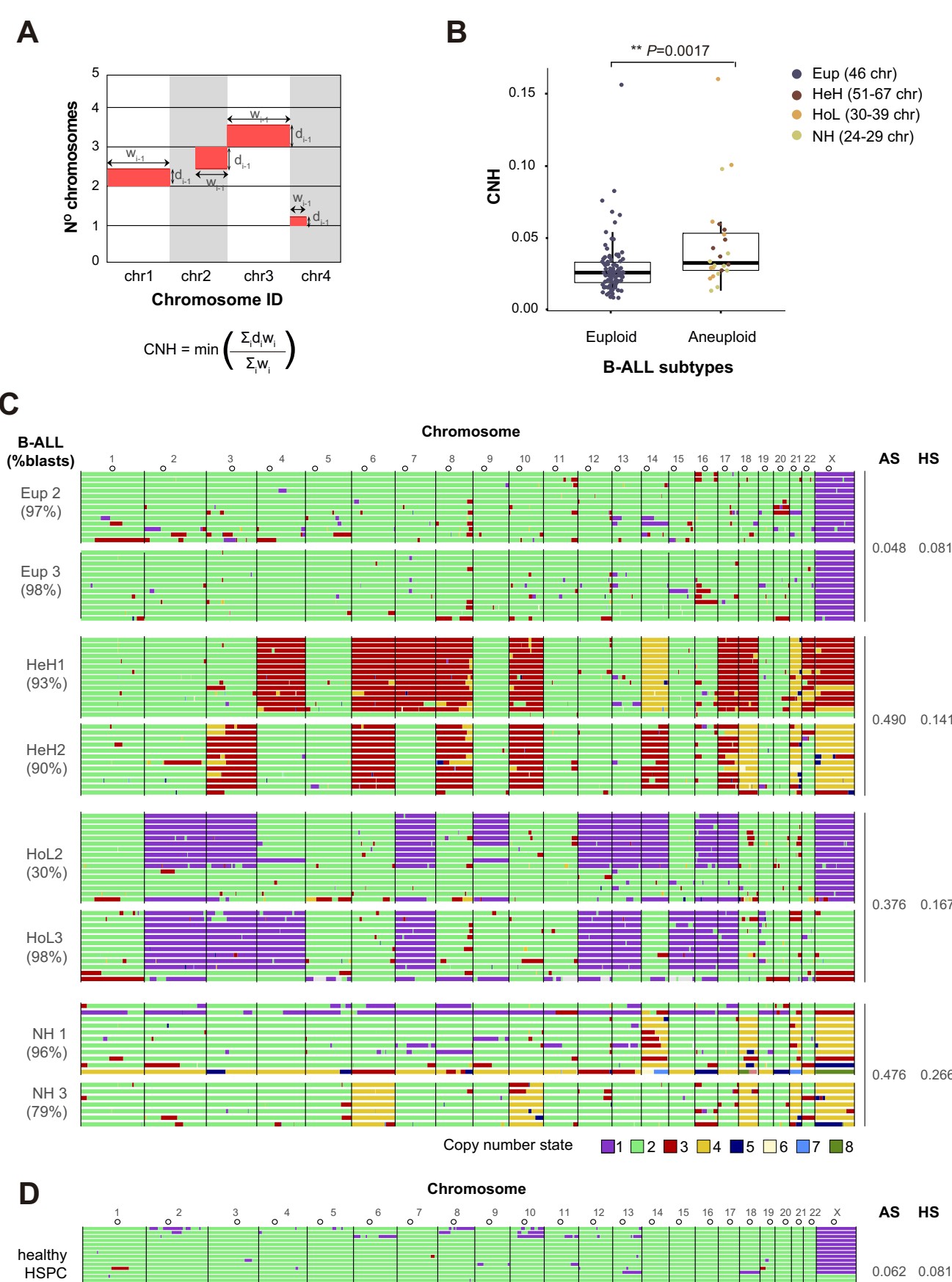

**Figure 1. Aneuploid cB-ALL subtypes consistently show high chr-CNH.**

(A) Strategy to calculate the chr-CNH described by van Dijk et al (2021). (B) Box-plot showing the chr-CNH calculated as in (A) in the indicated samples ($n = 118$ samples; $n = 94$ euploid and 24 aneuploid). The box begins in the first quartile (percentile 25%) and ends in the third quartile (percentile 75%), central horizontal line represents the median value. Lines represent segments of furthest data without accounting for outliers. Unpaired Student's t-test was used. **$P < 0.01$. (C, D) scWGSeq (bin size 1 MB) of cB-ALL primary BM samples (C) and healthy CD34+ HSPCs (D). Single cells are represented in rows and chromosomes are plotted as columns. Copy-number states are plotted using the Aneufinder algorithm. Aneuploidy and heterogeneity scores (AS and HS, respectively) in each cB-ALL group are shown on the right. Source data are available online for this figure.

**Table 1. Cytogenetic-molecular and biological data of cB-ALL patients used to generate patient-derived xenograft (PDX) models.**

| ID | CYTOGENETICS | FISH/Molecular biology | Gender | Age | % blasts | MRD | Relapse | DFS | Death |
|---|---|---|---|---|---|---|---|---|---|
| Eupl1 | 46,XX[20] | – | F | 5 | 100 | Neg | No | 10 | No |
| Eupl2 | 46,XY[21] | nuc ish(ABL1,BCR,MLL,ETV6,AML1,E2A,IGH)x2[200] | M | 5 | 97 | Neg | No | 5 | No |
| Eupl3 | 46,XY[20] | nuc ish(BCR-ABL,MLL,ETV6/RUNX1)x2[200]/DNi EUPLOID | M | 11 | 98.2 | Neg | No | 6 | No |
| HeH1 | 56–59,XX,+X,+4,+5,+6,+7,+8,+10,+14,+14,+14,+15,+17,+18,+21,+21,+22[cp6] | – | F | 4 | 93 | Neg | No | 7 | No |
| HeH2 | 57–58,XX,+X+X,+3,+6,+8,+8,+10,+10,+13,+14,+14,+17,+18,+18,+21,+21[cp13]/46,XX[6] | – | F | 5 | 90 | Neg | No | 5 | No |
| HeH3 | 54, XX,+X,+6,+8,+14,+17,+18,+21,+21[15]/54,idem,−13,+mar[11]/46,XX[19] | – | F | 4 | 100[a] | Pos | No | 5.8 | No |
| HoL1 | 46,XX[18]/32–38,XX,−3,−5,−6,−7,−10,−13,−18,−20[6]/50–54,XX,+17,+18,inc[4] | nuc ish(RP11-705O1x3),(RARAx3)[18/100] | F | 3 | 90 | Neg | No | 7 | No |
| HoL2 | na | nuc ish(MLLx1-3,BCR-ABLx1-3,ETV6/RUNX1 × 1–4)[200]/DNi=0.76(HoL) | F | 11 | 30 | Neg | No | 6 | No |
| HoL3 | 37,XX,−2,−3,−4,−7,−12,−13,−15,−16,−17[6]/73,XXX,+X,+1,−2,−3,−4,+5,+6,−7,+8,+9,+11,−12,−13,+14,−15,−16,−17,+18,+19,+20,+21,+22[8] | – | F | 13 | 98 | Pos | Yes | 0.43 | Yes |
| NH1 | 27<1n>,XX,+14,+18,+21[20] | – | F | 14 | 97 | Pos | Yes | 0.27 | Yes |
| NH2 | 52,XY,+X,+Y,+14,+14,+21,+21[8]/46,XY[2] | Masked near haploidy (Promega Powerplex 16) | M | 4 | 97.5 | Neg | No | 9.84 | No |
| NH3 | 46,XY[20] | nuc ish(2,3,5,7,8,9,11,12,13,14,15,16,17,22)x1-2[200] | M | 16 | 78.9 | Neg | No | 8.99 | No |

Eup (Euploid): 46 chr.; HeH (High hyperdiploid): 51–67 chr.; HoL (Low hypodiploid): 30–39 chr.; NH (Near haploid): 24–29 chr.
*DFS* disease-free survival (years), *DNi* DNA index, *F* female, *M* male, *MRD* minimal residual disease status post-induction, *na* not assessed, *nuc ish* interphase nuclei FISH, (-) no data available.
[a]FACS-sorted sample.

common in NH-B-ALL PDX samples (Fig. EV1B), suggesting that different causes predominantly underlie CIN in different aneuploid cB-ALL subtypes.

As the rates of specific mitotic defects are highly influenced by the rates of specific mitotic phases (*i.e.*, misaligned chromosomes are observed in early mitosis), we next analyzed the rates of chromosome mis-segregation, observed as lagging chromosomes or bulky anaphase bridges in late mitosis, to define the actual CIN levels in the PDX samples. As expected, chromosome mis-segregation levels were significantly higher across aneuploid than in euploid B-ALL-PDX samples ($P < 0.001$; Figs. 3E and EV1C), indicating higher CIN levels in the former. We noted a low variability in the rates of mitotic errors in independent PDXs derived from the same primary cB-ALL sample, confirming the validity of the CIN levels observed (Fig. EV1A, C). Collectively, our data confirm variable levels of ongoing CIN in aneuploid cB-ALL-PDX samples. The different types of mitotic defects observed across aneuploid cB-ALL samples suggest defects in different components of the mitotic machinery as underlying causes of CIN.

## CIN is associated with moderate levels of intraclonal chr-CNH in aneuploid cB-ALL

PDX models provide a unique opportunity to directly test how ongoing CIN contributes to cell-to-cell genomic heterogeneity. Having established that CIN is widespread in aneuploid cB-ALL-PDX models, we next determined the rates of chr-CNH present in these samples using Multicolor FISH (M-FISH) on freshly-isolated blasts (Fig. 4A). As expected, normal karyotypes with very low chr-CNH were detected in all Eup-B-ALL PDX samples, as observed by a homogeneous karyotype heterogeneity score (kHS) of 0.99 (Figs. 4B and EV1D). Contrastingly, modal chromosome numbers were highly variable across the aneuploid cB-ALL-PDX samples, ranging from 27 to 58 chromosomes (Fig. EV1D). Notably, modal chromosome numbers in the hyperdiploid range were detected in all HoL- and NH-B-ALL samples, suggesting that endoreduplicated hypodiploid clones become dominant after PDX expansion, by drift or by specific selective pressure (Fig. EV1D). Ploidy shifts in HoL- and NH-B-ALL samples were clearly observed when comparing the modal chromosome numbers between the primary cB-ALL samples

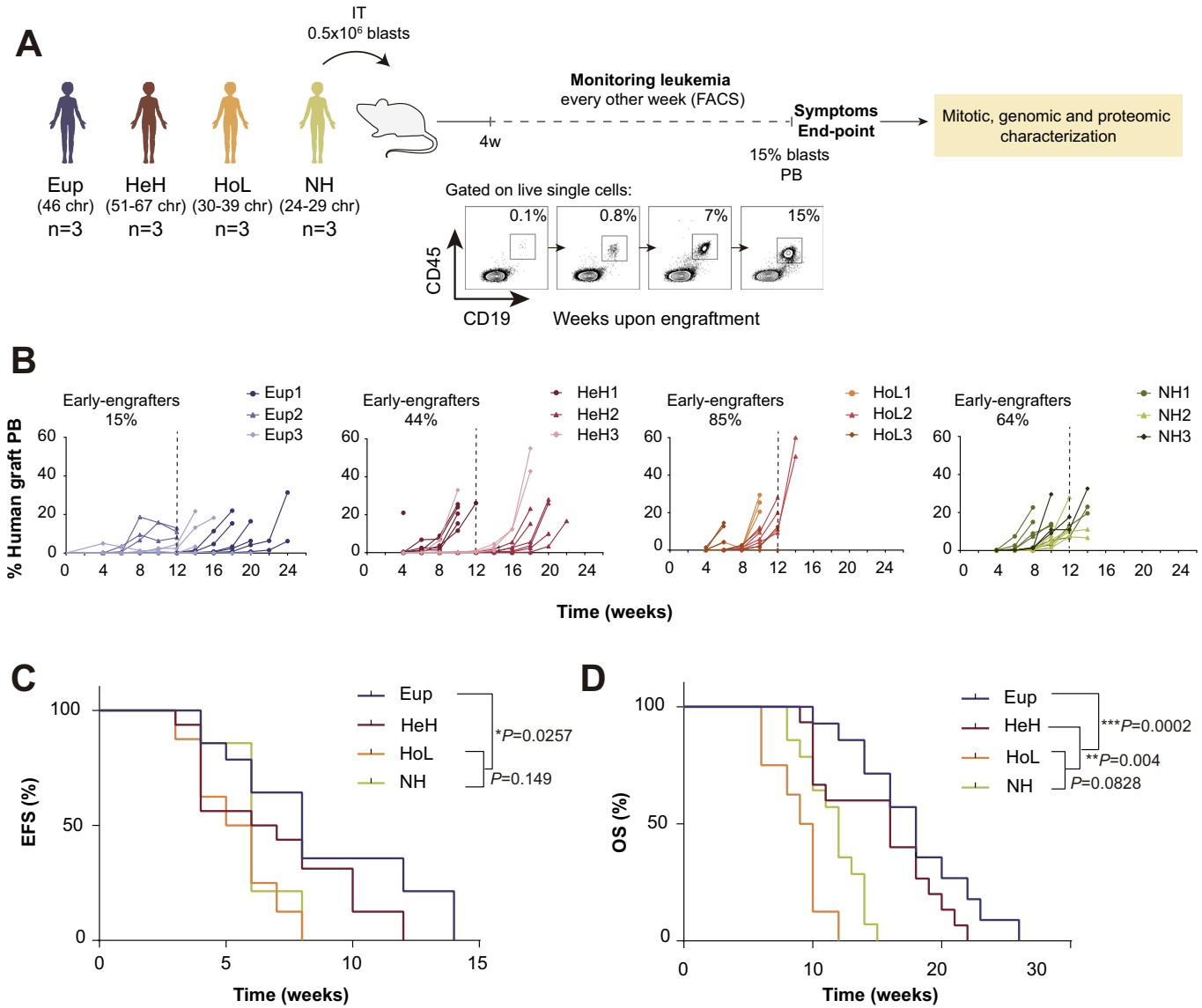

**Figure 2. PDXs models recapitulate the clinical outcome of cB-ALL patients.**

(A) Experimental design to generate and follow-up PDX models derived from primary cB-ALL samples ($n = 56$ PDX from 12 B-ALL samples, 3 samples per cB-ALL ploidy group). Representative images of leukemia monitoring in PB by flow-cytometry (bottom). (B) cB-ALL follow-up in PB of individual PDXs with the indicated cB-ALL samples. Each graph represents mice from the indicated ploidy groups ($n = 13$ Eup-, 16 HeH-, 13 HoL- and 14 NH-B-ALL). (C, D) Kaplan–Meier survival curves of EFS (C) and OS (D) of the indicated groups of PDX-derived cB-ALL samples. Statistical significance was determined by Log-rank Mantel-Cox tests. $*P < 0.05$, $**P < 0.01$, $***P < 0.001$. Data information: The experiments represent a minimum of three PDX samples (technical replicates) per cB-ALL sample (biological replicates) in each ploidy group. Source data are available online for this figure.

and their resulting PDXs (Fig. EV1E). The extent of karyotype heterogeneity was moderately higher across aneuploid cB-ALL than in Eup-B-ALL PDX samples, with the kHS ranging from 1.10 to 1.32 ($P < 0.001$; Fig. 4B,C). Of note, modal karyotypes in the PDX cells matched those of the primary cB-ALL cells or showed exact doubled-up karyotypes in the case of HoL- and NH-B-ALL (Fig. EV1F and Appendix Table S1), indicating that rates of CIN in cB-ALL cells are reflected as intraclonal genomic variability within an already adapted karyotype, which may favor the survival and proliferation.

Chr-CNH was not equally distributed across the karyotype, but specific chromosomes showed higher rates of variability, as observed by chromosome copy-numbers with standard deviation (SD) ≥ 1, such as chromosomes 6, 14, 17, 18, 21, and X in HeH-B-ALL; chromosomes 1, 6, 11, 8, 10, 18, 20, 21, and X in HoL-B-ALL; and chromosomes 14, 18, 21, and X in NH-B-ALL (Fig. EV1G). Notably, the most variable chromosomes were coincident with those gained or retained as disomies in the corresponding primary samples, suggesting that chromosomal gains at leukemia initiation buffer the negative consequences of CIN on cellular fitness,

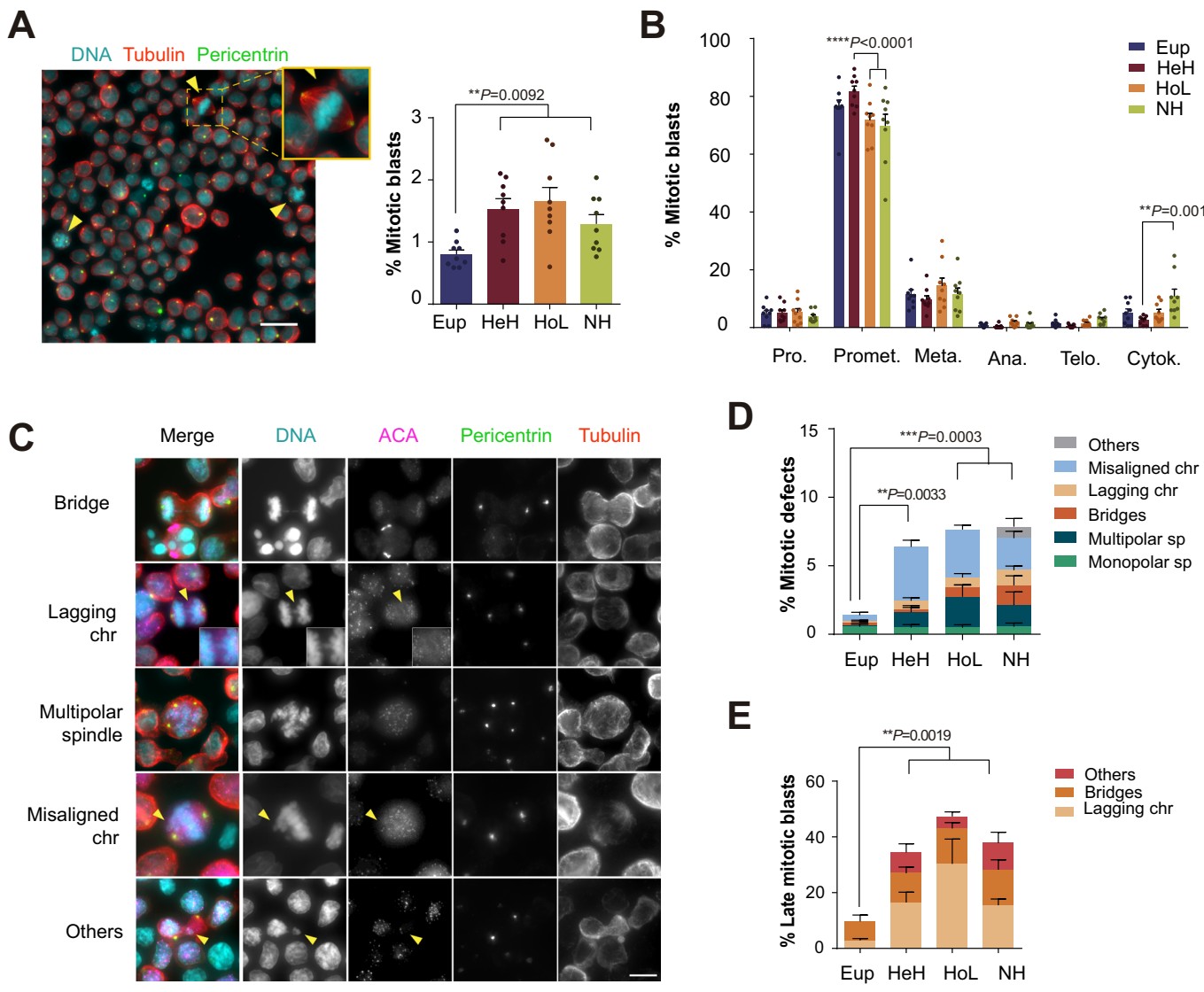

**Figure 3. Aneuploid cB-ALL show higher rates of mitotic defects in PDX models.**

(A) Analysis of the mitotic index in cB-ALL PDX models by immunofluorescence (IF) analysis. Left, representative DNA-tubulin-pericentrin IF staining of cB-ALL-PDX cells. Yellow arrowheads point to mitotic cells in the selected field. Right, percentage of mitotic cells in the indicated cB-ALL ploidy group ($n = 36$, 3 cB-ALL-PDX samples per group; Eup = 20,750, HeH = 19,480, HoL = 22,274, NH = 22,183 cells). Scale bar: 10 μm. (B) Mitosis progression of cB-ALL-PDX cells from (A). Graph shows the frequency of mitotic cells at the indicated mitotic phases. (C) Representative DNA-Centromere (ACA)-tubulin-pericentrin IF staining of cB-ALL-PDX cells identifying the indicated mitotic defects. Yellow arrows point to the indicated mitotic defects. Scale bar: 10 μm. (D) Frequency of mitotic cB-ALL-PDX-expanded primary blasts with the indicated mitotic defects (36 PDX samples, 3 PDX per primary sample; $n = 200$ mitotic cells per PDX sample, Total=7200 mitotic cells). (E) Frequency of cB-ALL-PDX-expanded primary blasts at late mitosis with the indicated chromosome segregation defects (36 PDX samples, 3 PDX per primary sample; $n = 123$ Eup, 116 HeH, 163 HoL, and 186 NH late mitoses). Data information: Data are presented as mean ± SEM. Statistical significance in (A), (B), and (D) was determined by two-way ANOVA test, and by Wilcoxon rank-sum test in (E). **$P < 0.01$, ***$P < 0.001$. The experiments represent a minimum of three PDX samples (technical replicates) per cB-ALL sample (biological replicates) in each ploidy group. Source data are available online for this figure.

allowing certain karyotype variation that may ultimately be involved in clonal adaptation.

A direct relationship has been established between CIN and Chr-CNH in different organisms and tumor samples (Bolhaqueiro et al, 2019; Bollen et al, 2021; Foijer et al, 2014). To test the association of CIN rates with chr-CNH in cB-ALL, we correlated the different parameters analyzed in cB-ALL mitotic blasts, including mitotic index, mitotic defects, and chromosome mis-segregation rates, with the reported kHS by M-FISH. Strongly

significant positive correlations were observed for all these parameters with kHS in cB-ALL-PDX samples (Fig. 4D), providing a direct link between CIN and chr-CNH in cB-ALL-PDX models. The observed non-clonal variability supports a punctuated evolution model in cB-ALL, where one or a few dominant clones stably expand during leukemogenesis (Davis et al, 2017), with intraclonal karyotype variations shaping the chr-CNH spectrum, as was previously observed in a mouse model for T-ALL (Shoshani et al, 2021).

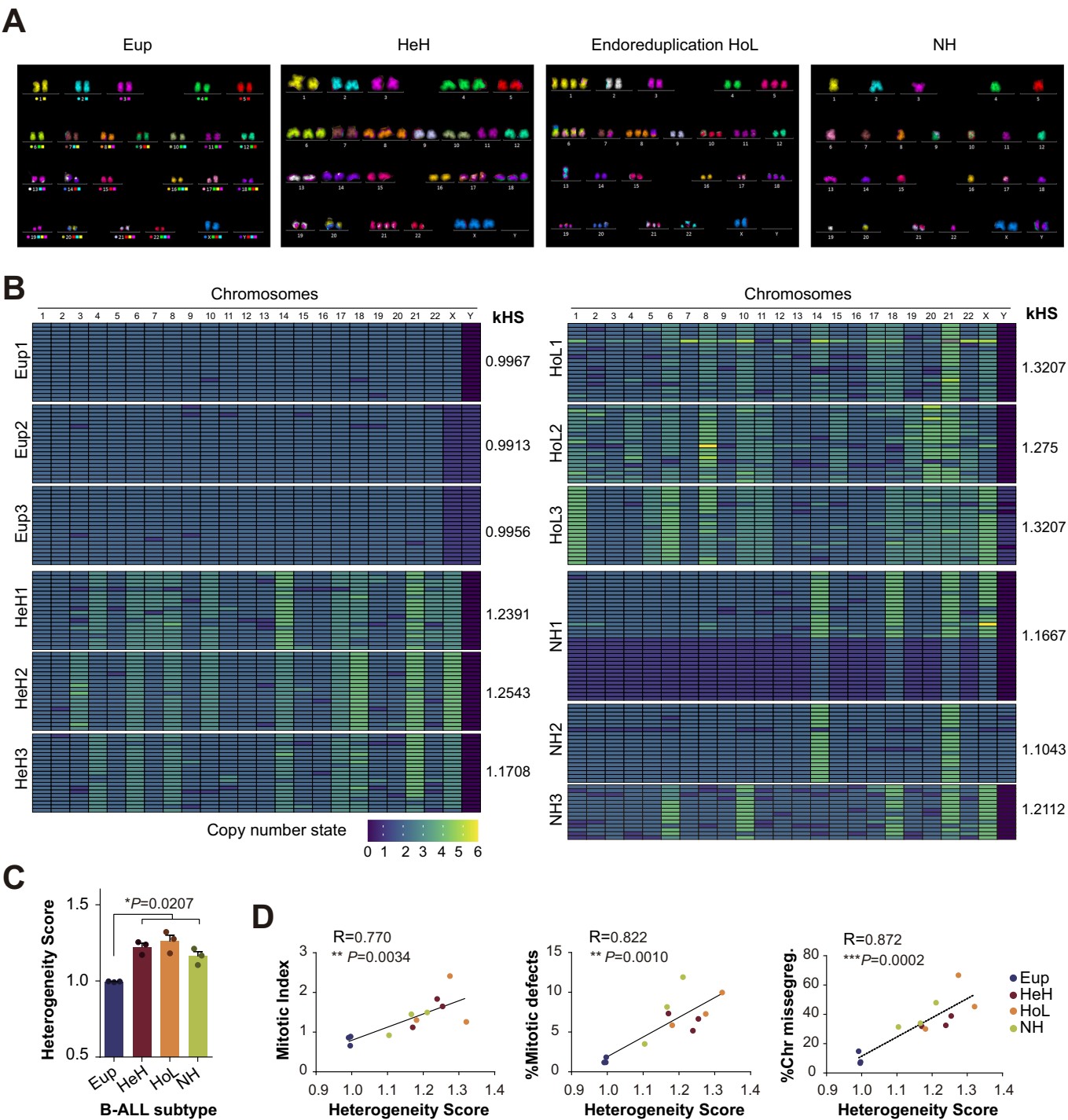

**Figure 4. High rates of mitotic defects underlie large chr-CNH in PDX models of aneuploid cB-ALL.**

(A) Representative M-FISH karyotypes from cB-ALL-PDX-expanded primary blasts of different cB-ALL subgroups. Chromosome-specific fluorophore combinations are indicated at each chromosome pair in the karyogram at the left. (B) Heatmaps showing the M-FISH results obtained for the indicated samples ($n = 247$ metaphases; $n = 20$ for each sample except for NH#1 ($n = 33$) and NH#3 ($n = 14$)). Single cells are represented in rows and chromosomes are plotted as columns. Copy-number states are plotted as depicted at the *bottom*. Heterogeneity scores (kHS) for each sample are indicated in the right. (C) HS obtained from (B) in the indicated ploidy groups ($n = 12$; 3 Eup-, 3 HeH-, 3 HoL- and 3 NH-B-ALL). Graphs represent the mean value of each ploidy group and error bars represent the SEM. Holm-Sidak's multiple comparison test was used. *$P < 0.05$. (D) Correlations between HS obtain from (B) and mitotic index (*left*), rates of mitotic defects (*middle*), and rates of chromosome mis-segregation (*right*) in the indicated B-ALL ploidy groups. Peason's correlation tests were used. **$P < 0.01$, ***$P < 0.001$. Data information: The experiments represent three PDX samples (technical replicates) per cB-ALL sample (biological replicates) in each ploidy group. Source data are available online for this figure.

## CIN influences cB-ALL progression in PDX models

CIN has been associated with tumor progression across different cancer types (Bakhoum et al, 2018; Foijer et al, 2014; Godek et al, 2016; Hoevenaar et al, 2020; van Dijk et al, 2021). To test whether this association holds true in cB-ALL, we next assessed the relationship between the levels of different CIN parameters, including mitotic index, rates of mitotic defects, rates of chromosome mis-segregation and chr-CNH, with the leukemia kinetics observed by PB graft monitoring in the cB-ALL-PDX models. Two stages were differentiated to assess in vivo cB-ALL kinetics (Fig. 5A): first, the time-to-engraft was determined as the time from transplant to the time when cB-ALL engraftment was first detectable (>0.1%) in PB; second, time-of-disease progression was determined as the time elapsed from first detectable cB-ALL engraftment until overt leukemia (defined as >15% blasts in PB, which represents >80% blasts in BM) (Molina et al, 2020). Results showed no correlation between any CIN parameter with the time-to-engraft in mice (Fig. 5B). However, significant correlations were observed between the rates of mitotic defects and chromosome mis-segregation with disease progression (Fig. 5C). A trend for significance between kHS and disease progression was also observed, suggesting that other factors beyond CIN are involved in chr-CNH and disease progression in cB-ALL (van den Bosch et al, 2022).

To further examine the relationship between CIN and cB-ALL disease progression, we classified cB-ALL-PDX samples according to their rates of mitotic defects into three CIN groups: (i) CIN$^{low}$, within the percentile 0 and 25, with mitotic defects ranging from 0.5% to 2.75%; (ii) CIN$^{mid}$, within the percentile 25 and 75, with mitotic defects ranging from 2.75% to 7.3%; and (iii) CIN$^{high}$, within the percentile 75 and 100, with mitotic defects ≥7.3% (Fig. 5D,E). Results revealed reduced OS rates with increasing levels of CIN within our cohort of cB-ALL PDX samples (Fig. 5F), further demonstrating the relationship between CIN and disease progression in cB-ALL.

## Mathematical modeling confirms a relationship between CIN and disease progression

We further explored the role of CIN in the progression of cB-ALL by using an in silico approach to predict the dynamics and karyotype evolution of a leukemic population in the presence of different levels of CIN. Building this discrete model at a single-cell resolution required tracking each resulting karyotype resulting from cell division to study whether specific karyotypes are preferentially selected. To do this, we determined the cell fitness (Φ) and its interaction with variable selection patterns (σ), which are key factors in shaping the resulting karyotypes selected over time (Fig. 6A). We assigned a Φ value based on features that influence leukemogenesis and tumor development (α, β, γ; ω$_1$), taking into account structural properties affecting chromosome segregation (ω$_2$), such as the centromeric size (S$^{Cen}$) and total gene density (S$^{GD}$) (Fig. 6B; Appendix Tables S2 and S3) (Lynch et al, 2022). Since the relative contribution to Φ, or weight (ω), of each of these processes is largely unknown and to prevent potential conceptual biases, we assumed an equal contribution from both types of features (ω$_1$ = ω$_2$ = 1), and in turn the same contribution of the sub-features.

We aimed to test the contribution of ongoing CIN in aneuploid cB-ALL, focusing on the impact of CIN in aneuploid conditions rather that the mechanisms leading to the aneuploid karyotypes. For this purpose, we set aneuploid leukemic cells as the starting point for our simulations (Fig. 6A). Even in the absence of CIN ($P_{CIN} = 0$), we simulated an actively proliferating leukemic population (Fig. 6C). At higher levels of CIN, the cell growth slowed down, as evidenced by the delay in the time taken to reach the carrying capacity threshold set at $1 \times 10^6$ cells (47 days at $P_{CIN} = 0$, delayed to 107 days at $P_{CIN} = 0.3$) (Figs. 6C and EV2A). In the absence of selective pressures (σ) associated with CIN ($P_{CIN} = 0$), the cell division rate ($P_{DIV}$) and average Φ remained almost constant throughout time (Fig. 6D,E). However, even though initial growth was compromised, we observed an increase in proliferation capacity and Φ values with variable levels of CIN (Fig. 6C–E), suggesting a positive correlation between CIN and cell proliferation rates upon adaptation to the negative impact on cellular fitness. Interestingly, the consolidation time required to reach the maximum $P_{DIV}$ was shorter for low-to-mid CIN than for high levels of CIN (Fig. EV2B). Therefore, our simulation identified a range of optimal values for low-to-mid CIN, corroborating the paradox of CIN in cancer progression (Vasudevan et al, 2021). Importantly, the optimal CIN levels identified in these simulations align with experimentally observed rates of mitotic defects (ranging from 0% to 17%, Figs. 3C–E and EV2A–C).

To evaluate the effect of CIN on chr-CNH, we tracked the genomic heterogeneity by registering individual cell karyotypes over time. Our results showed that karyotype heterogeneity was initially greater at low-to-mid CIN levels but decreased by the end of the simulation (Fig. 6F). This suggests that CIN allows cells to modify their proliferative potential ($P_{CIN}$) and to explore the space of viable karyotypes without significantly compromising their viability. Consequently, it results in more stable cell populations comprising a subset of adapted karyotypes. Notably, our modeling approach predicts that exceeding high CIN levels drive cells towards a decreased karyotype heterogeneity compared to low-to-mid CIN levels (max karyotype diversity of 4.7 in $P_{CIN} = 0.05$ and 3.1 in $P_{CIN} = 0.4$; Fig. 6F). Furthermore, maximum karyotype diversity is delayed as CIN levels increase (Fig. EV2B). Altogether, high CIN levels give rise to karyotypes that more frequently compromise cell viability more frequently, thereby decreasing cell fitness (Fig. 6E) and subsequently reducing karyotype heterogeneity rates (Fig. 6F).

Having established a robust simulation approach to assess the role of CIN in cB-ALL progression, we next performed simulations using karyotypes observed in two primary aneuploid cB-ALL samples from our cohort (HeH1 and NH1). Consistent with previous data, the results showed increased Φ and $P_{DIV}$ at low-to-mid CIN levels (Fig. EV2C,D). Despite the predicted increase in karyotype heterogeneity at later simulation times, the modal karyotypes were preserved for intermediate CIN values for both samples (Fig. 6G), suggesting intraclonal karyotype heterogeneity, as observed experimentally (Figs. 4B and EV2F). Remarkably, hyperdiploid karyotypes were positively selected from the NH karyotype in NH1 (Fig. EV2F), mirroring our experimental findings in both primary and PDX samples. In the NH1 sample, an increased tolerance to CIN was observed in silico, as extra generations were required for these cells to reach viability-

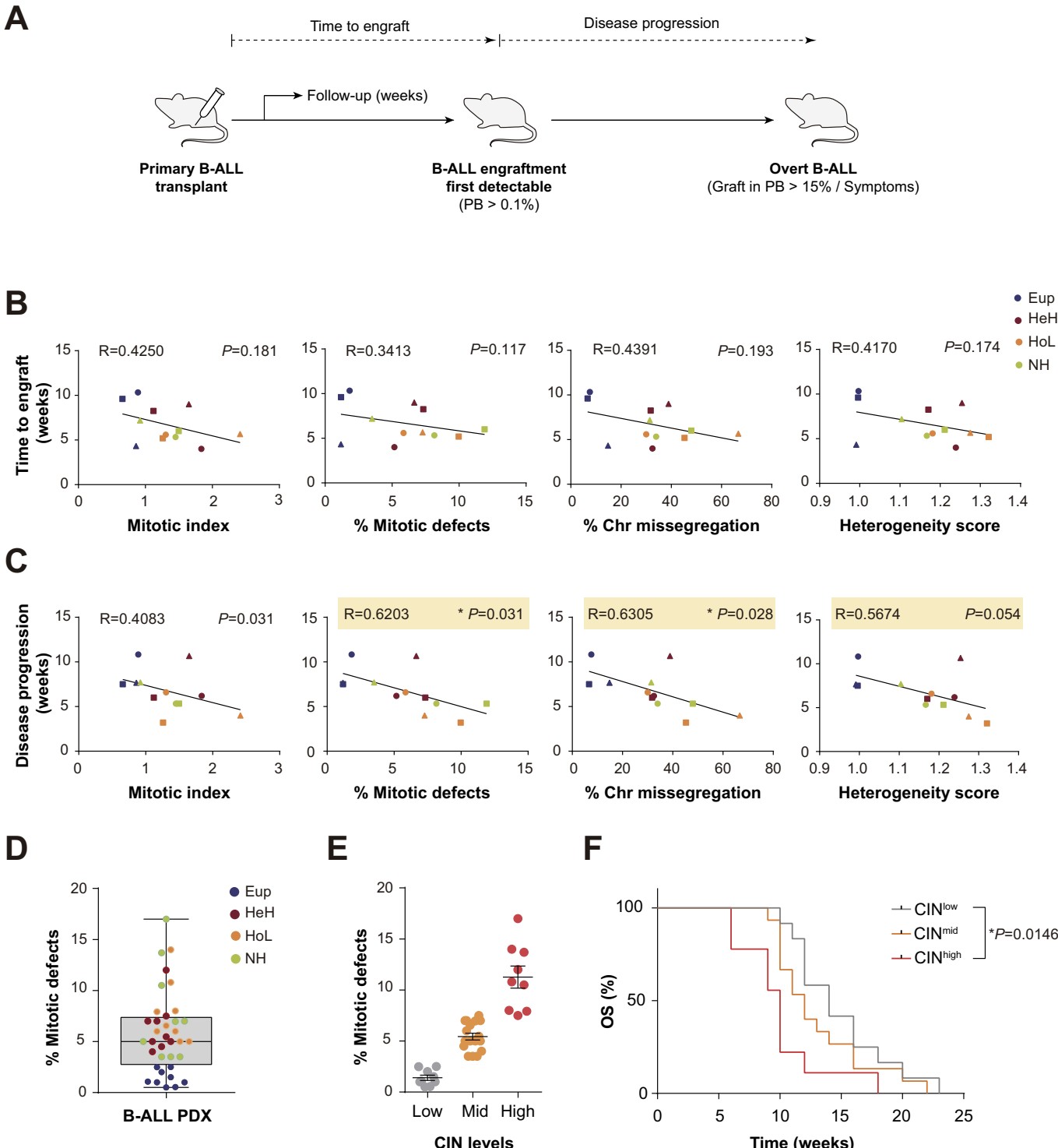

threatening karyotypes. Slightly increased average cell fitness in the absence of CIN ($P_{CIN} = 0$) was observed in NH1 simulations due to whole-genome duplication. It is important to note that our focus should not be solely on increased fitness or average proliferation rate, but also on whether the karyotypes obtained at the end of the simulation align with those observed in real samples. As we increase the CIN level, aberrant chromosomal gains and losses

accumulate, including unobserved monosomies in real cases, which further underscores that intermediate-low CIN levels appear to be optimal. Collectively, the mathematical and experimental data are consistent and suggest that low-to-mid CIN levels allow aneuploid B-ALL cells to explore a broad spectrum of phenotypic-karyotypic states, thus increasing their adaptive potential and proliferation rate. This acts as a mechanism to reach a certain level of karyotypic

**Figure 5.   CIN is associated with disease progression in PDX models of cB-ALL.**

(A) Experimental design to assess variables of clinical outcome in PDX models of cB-ALL (time-to-engraft and time-to-disease progression). (B, C) Correlations between time-to-engraft (B) and time-to-disease progression (C) and mitotic index, rates of mitotic defects, rates of chromosome mis-segregation, and heterogeneity score (kHS) in the indicated B-ALL ploidy groups. Graphs represent the average time-to-engraft and time-to-disease progression from the PDX samples in each primary cB-ALL, represented by colored symbols (circles, triangles and squares identify independent primary cB-ALL). Peason's correlation tests were used. *$P < 0.05$. (D) Box-plot showing the rates of mitotic defects in cB-ALL-PDX from Fig. 3. Each dot represents a cB-ALL-PDX sample from the indicated ploidy group. Box represents the quartiles 25–75 and horizontal line represents the mean value, error bars represent the SD ($n = 36$). The box begins in the first quartile (percentile 25%) and ends in the third quartile (percentile 75%), central horizontal line represents the median value. Vertical line represents segment of furthest data from minimum (bottom) to maximum (top) values. (E) Rates of mitotic defects in the indicated cB-ALL-PDX groups. Horizontal lines show the median and error bars represent the SEM. (F) Kaplan–Meier plot showing the OS of cB-ALL-PDX from the indicated groups. Statistical significance was determined by Log-rank Mantel-Cox tests. *$P < 0.05$. Data information: The experiments represent a minimum of three PDX samples (technical replicates) per cB-ALL sample (biological replicates) in each ploidy group. >Source data are available online for this figure.

heterogeneity (Fig. 6H) without compromising cell viability. Intrinsic and extrinsic cellular selective pressures, such as those imposed by cell-to-cell competition and the harsh environment during leukemia progression, are key factors in determining their advantageous potential. While these factors have not been explicitly included in our computational framework, they would confer a selective advantage to the more heterogeneous and adaptable population while penalizing those with excessively high genome instability. In summary, CIN operates as a source of karyotypic and cellular diversity.

## Adaption of the mitotic spindle machinery to regulate chromosome segregation is associated with CIN in cB-ALL

Chromosome mis-segregation leads to a myriad of cellular and metabolic stresses which result in a strong negative selection when occurring in diploid cell types, but which are tolerated in aneuploid cancer cells (Chunduri and Storchova, 2019; Crasta et al, 2012; Donnelly et al, 2014; Stingele et al, 2012; Torres et al, 2007). Cancer cells undergo adaptive resistance rooted from variable cellular signaling networks that involve feedback-dependent homeostatic control. This includes changes to the signaling networks that regulate chromosome segregation fidelity during mitosis to maintain CIN levels under a viable threshold (Bakhoum et al, 2018; Orr et al, 2016; Sansregret et al, 2017). To examine the cellular signaling networks involved in the adaptive resistance to CIN in aneuploid cB-ALL cells, we performed mass-spectrometry (MS) analyses using whole-cell lysates from all our cB-ALL PDX samples and correlated absolute protein abundances with the rates of mitotic defects in these samples (Fig. 7A). Our results showed a total of 213 and 226 proteins positively and negatively correlated with CIN ($P < 0.05$), respectively (Figs. 7B and EV3). Gene ontology (GO) analyses with the positively-correlated proteins revealed a strong association with mitosis regulatory pathways, including establishment and maintenance of cytoskeleton polarity and regulation of chromosome segregation, and with ncRNA processing (Fig. 7C). Contrastingly, negatively-correlated proteins were mainly associated with chromatin regulation, metabolic function, and integrin-mediated cell signaling (Fig. 7C). The relevance of these pathways was confirmed by protein–protein interaction network analysis. Two discrete protein-interaction clusters were observed with the positively-correlated proteins, consisting of proteins associated with mitosis regulation and ncRNA processing (Fig. EV4A). Protein interaction network analyses with negatively-regulated proteins identified clusters associated with the

mitochondrial electron transport chain, chromatin organization, and integrin-mediated cell signaling (Fig. EV4B).

The positive correlation between mitosis regulatory factors and CIN suggests adaptation mechanisms to the mitotic stresses imposed by CIN in aneuploid cB-ALL cells. To test the possibility that defects in the mitotic machinery in aneuploid cB-ALL underlie the positive correlation between mitotic spindle and chromosome segregation pathways, we analyzed St Jude's hospital RNAseq data from 765 patients with cB-ALL to investigate the differential transcriptomic signatures between aneuploid and euploid cB-ALL samples (Fig. EV5) (Gu et al, 2019). Of note, patients with aneuploid cB-ALL, except for patients with HoL-B-ALL mostly diagnosed with Li-Fraumeni syndrome harboring TP53 mutations (Molina et al, 2021b), clustered together in Uniform Manifold Approximation and Projection (UMAP) analyses (Fig. 7D), suggesting that, irrespective of their clinical outcome, aneuploidy makes a significant contribution to the transcriptomic signature of aneuploid subtypes of cB-ALL. Importantly, GO analyses with the differentially-expressed genes between aneuploid and euploid cB-ALL revealed a down-regulation of genes associated with the mitotic spindle (Fig. 7E,F and Dataset EV2). Hence, the positive correlation of mitotic spindle maintenance factors with CIN may play a key role in buffering the mitotic spindle defects observed in cB-ALL. Furthermore, the top upregulated pathway in aneuploid cB-ALL samples was IL6 JAK-STAT3 signaling (Fig. 7E,F and Dataset EV2), which has been recently involved in adaptation of breast cancer cells to CIN (Hong et al, 2022). Together, CIN in cB-ALL is associated with a specific phenotype, involving mitosis and chromosome segregation regulatory factors, which may act as an adaptation mechanism to limit the extend of mitotic stress under a viable and advantageous rate for aneuploid cB-ALL cells.

## Discussion

Ongoing CIN is a hallmark of cancer and has been observed in many solid tumors using a variety of complementary methodologies (Bakker et al, 2016; Bolhaqueiro et al, 2019; Bollen et al, 2021; Carter et al, 2006; Duijf et al, 2013; Hoevenaar et al, 2020; van Dijk et al, 2021; Vasudevan et al, 2020; Xu et al, 2021). However, the presence and active role of CIN in cB-ALL is under debate and has been limited to studies of karyotype heterogeneity because primary leukemic cells fail to grow *ex vivo* (Alpar et al, 2014; Elghezal et al, 2001; Heerema et al, 2007; Paulsson et al, 2010; Ramos-Muntada et al, 2022; Talamo et al, 2010). A recent study applied scWGSeq to directly visualize the cell-to-cell variability of the entire karyotype

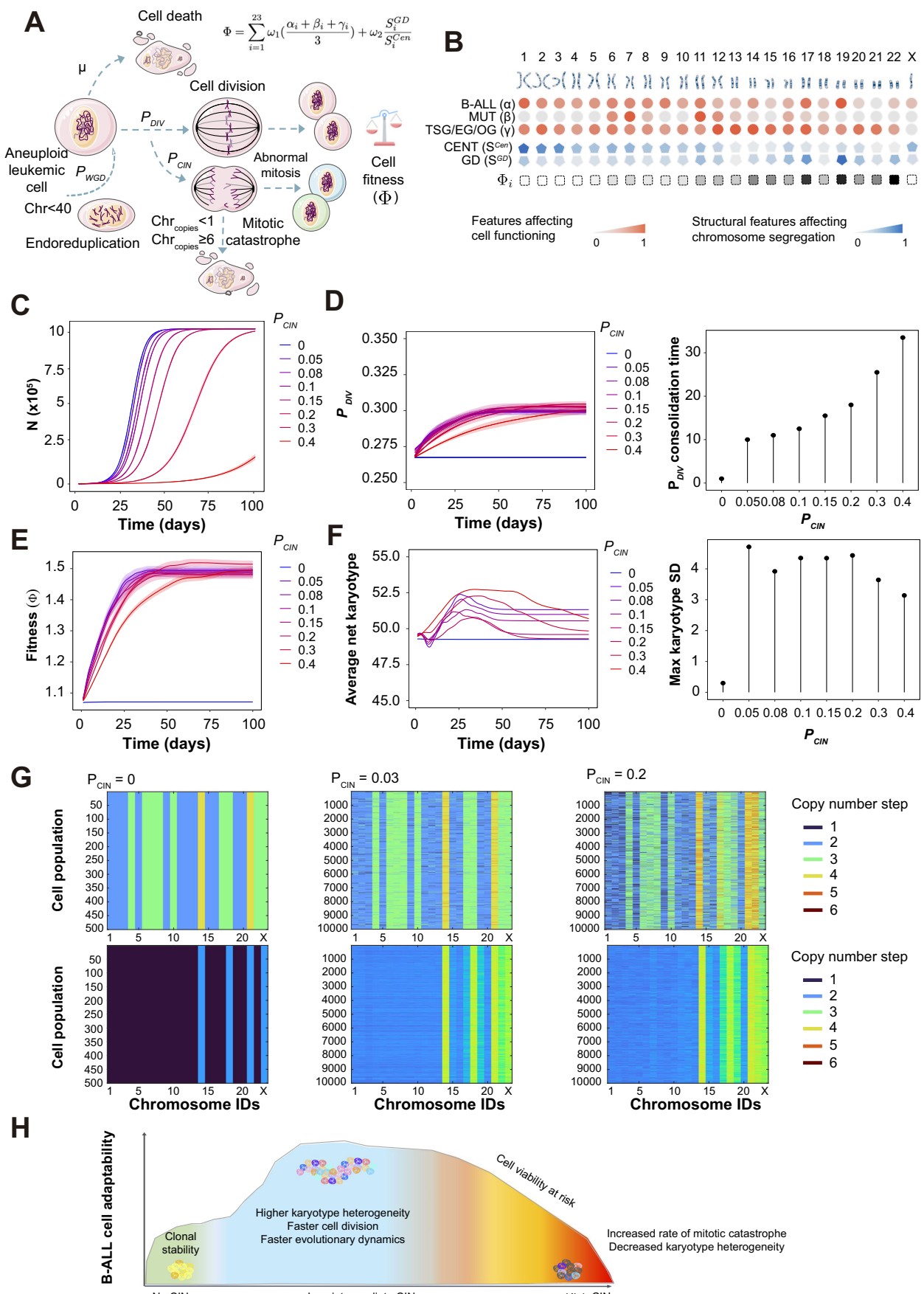

◄

**Figure 6.  Mathematical modeling suggests low-to-mid levels of CIN as drivers of clonal heterogeneity and disease progression in cB-ALL.**

(A) Schematic depicting the model parameters and cell processes considered in the algorithm used for in silico simulation of CIN. (B) Estimation of the contribution of each chromosome copy to global cell fitness, obtained by integrating functional and structural data for each single chromosome. Normalized density of genes associated with hematopoietic differentiation and cB-ALL (α), recurrent mutations in cB-ALL (β), tumor suppressor genes, essential genes and oncogenes (γ), centromere size ($S^{CENT}$) and chromosomal gene density ($S^{GD}$). (C–E) Simulations of cell number dynamics (C), cell division probability (D), and cell fitness (E) at the indicated CIN levels. Average fitness is plotted relative to that of euploid cells, with the latter set as 1. (F) Simulation of chromosomal clonal heterogeneity as depicted by the average net karyotype (left) and the variability of the net karyotype (right) at the indicated CIN levels. We conducted 50 independent simulations for each CIN level, and plotted the average curves. The SEM is represented through ribbons, except for karyotype standard deviation, which is depicted as a separate plot due to curve overlapping. (G) Simulations of the karyotype variability of samples HeH1 (top) and NH1 (bottom) at the indicated CIN levels. Copy-number states are shown as illustrated to the right. (H) Diagram summarizing the observations derived from in silico modeling portraying the relation between B-ALL cell adaptability and different levels of CIN. Source data are available online for this figure.

from nondividing cells in nine HeH-B-ALL samples, and found variable levels of genome-wide HS in the HeH-B-ALL samples, ranging from low-to-mid chr-CNH (Woodward et al, 2023). Here, we applied a novel method to infer chr-CNH from bulk WGS data using a large panel of cB-ALL samples (van Dijk et al, 2021) and scWGSeq to eight cB-ALL samples with different ploidies (Bakker et al, 2016). In line with previous studies (Woodward et al, 2023), both methods consistently showed higher levels of chr-CNH in aneuploid cB-ALL samples than in non-aneuploid cB-ALL samples. Our data also suggest the presence of ongoing CIN as an underlying cause of the variable chr-CNH observed across aneuploid cB-ALL samples.

CIN is defined by a persistent chromosome mis-segregation coupled to survival and propagation of aneuploid cells that acquired specific advantageous karyotypes (Godek and Compton, 2018). Accordingly, the assessment of chromosome segregation fidelity in actively dividing cancer cells is crucial to identify the presence of ongoing CIN. We have previously demonstrated the strong potential of PDX models to expand primary aneuploid and euploid leukemic cells in vivo and to characterize mitotic and chromosome segregation defects in dividing cells (Molina et al, 2020). Here, we expanded the use of these models by integrating complex cellular phenotypes with in vivo physiological data, which enabled a better comprehension of the levels of CIN and its effects on karyotype variability and leukemia progression in mice. Our results unequivocally show significantly higher rates of mitotic defects and chromosome mis-segregation in all aneuploid cB-ALL subtypes than in euploid cB-ALL. Although the CIN events detected in our assays may have various underlying causes, the most frequent mitotic errors were chromosome misalignments in the metaphase plate and multipolar spindles. Both defects are associated with the occurrence of erroneous kinetochore-microtubule attachments, more likely of the merotelic type, which are not detected by the SAC and ultimately lead to chromosome mis-segregation (Bakhoum et al, 2009; Thompson and Compton, 2011). The proliferative nature of the cB-ALL-PDX cells facilitated M-FISH karyotyping to assess chr-CNH. Consistently, a strong correlation between the rates of mitotic defects and chr-CNH was observed in our cB-ALL-PDX samples, indicating that ongoing CIN underlies, at least in part, karyotype heterogeneity in cB-ALL. Remarkably, as it was previously observed by scWGS, the levels of chr-CNH in cB-ALL-PDX were moderate and mainly characterized by intraclonal variability. Indeed, previous studies using single-cell sequencing and/or M-FISH identified surprisingly low-to-mid karyotypic variance in human tumors and cancer organoid models despite the presence of widespread CIN (Bolhaqueiro et al, 2019;

Nelson et al, 2020), supporting a relevant role of cell intrinsic and extrinsic factors in the selection of specific advantageous karyotypes generated by the ongoing CIN.

Noteworthy, hyperdiploid doubled-up clones were observed for most of the hypodiploid samples. This situation is observed in 60–65% of the HoL and NH-B-ALL patients with "masked hypodiploidy" (Carroll et al, 2019), and in cell lines generated from NH-B-ALL samples (NALM16 and MHH-CALL2) which show only a hyperdiploid clone consistent with an exact duplication of the original hypodiploid karyotype (Aburawi et al, 2011; Kohno et al, 1980). This indicates that doubled-up hypodiploid clones are positively selected in HoL- and NH-B-ALL samples. It is tempting to speculate that the endoreduplication of hypodiploid clones may buffer the effects of CIN events, increasing cell fitness by increasing the likelihood of acquiring adapted karyotypes under selective pressures. This hypothesis is supported by the fact that the only sample in our cohort with detectable near-haploid and doubled-up hyperdiploid clones (NH1) showed chr-CNH only in the hyperdiploid subpopulation.

CIN has been associated with tumorigenesis, therapeutic resistance, and poor survival outcomes in different human cancers (Bach et al, 2019; Duijf et al, 2013; Ippolito et al, 2021; Lukow et al, 2021; McClelland, 2017). We found that, indeed, this holds true in our cB-ALL PDX models, where the variable levels of ongoing CIN significantly associate with leukemia progression in mice. Of note, despite the cB-ALL cells were expanded in immune-deficient mice, which might impact their proliferation behavior, decreased OS rates were observed in samples with increasing CIN levels, suggesting that ongoing CIN may serve as a biomarker to improve risk stratification in patients with aneuploid cB-ALL. Recent evidence highlights the importance of CIN and related aneuploidies on tumor evolution. Higher levels of CIN suppress tumor growth when they surpass a critical threshold, likely due to aberrant gene and protein dosage stoichiometry (Donnelly et al, 2014; Oromendia et al, 2012). Furthermore, clinical outcomes have been shown to depend on the specific aneusomies produced by CIN, as the relative cell fitness is influenced by the expression of the chromosome-containing genes (Davoli et al, 2013; Ramos-Muntada et al, 2022; Sheltzer et al, 2017; Vasudevan et al, 2020). To account for these factors, we developed an in silico platform to determine the cell fitness after each virtual cell division and to infer the cell population growth and the karyotype evolution under different levels of CIN. Consistent with previous in silico simulations (Lynch et al, 2022), our results indicate increased cell proliferation rates and improved cell fitness when the levels of CIN remain below a critical threshold. Importantly, these levels correspond to the range of mitotic defects

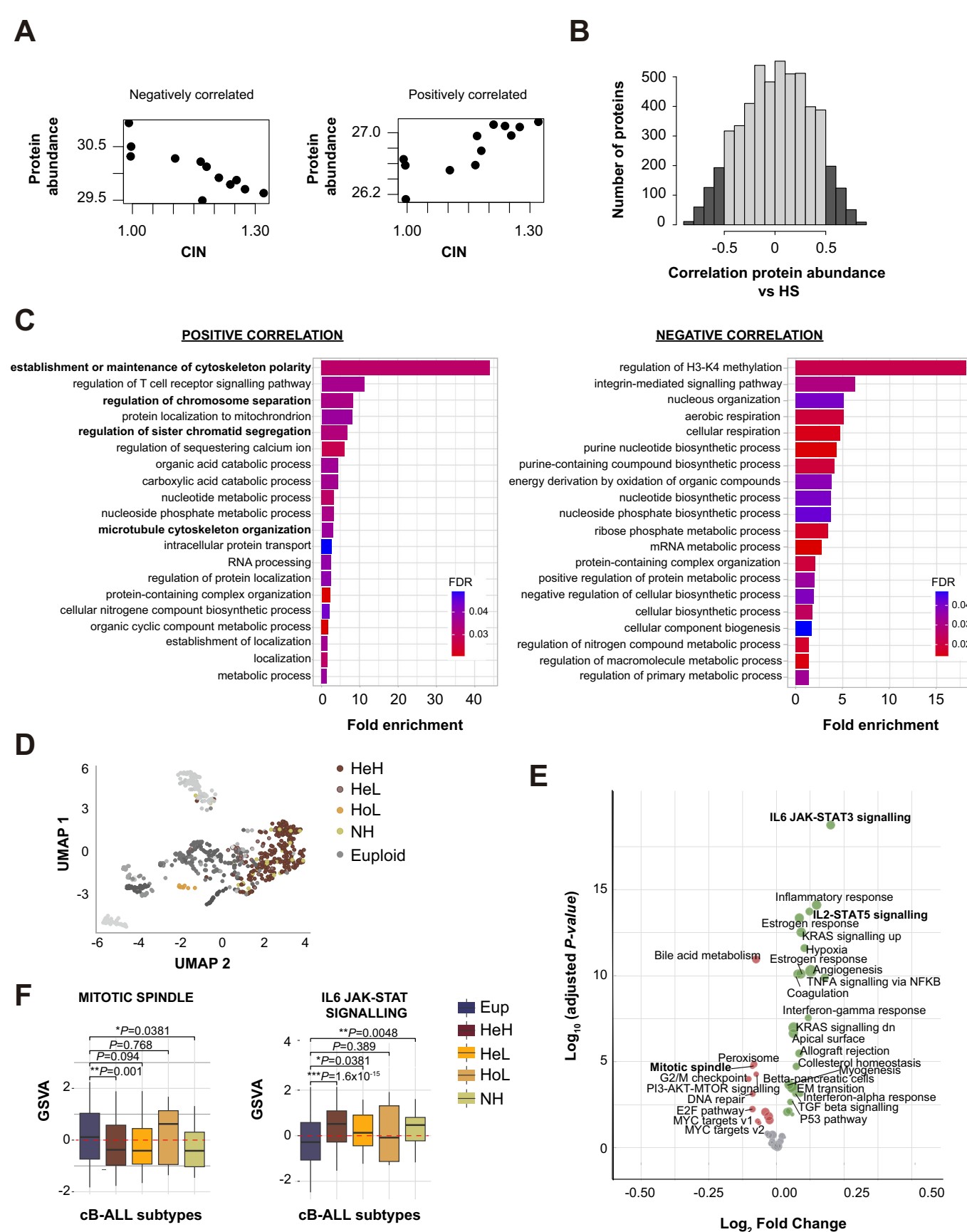

◀ **Figure 7. CIN in cB-ALL is characterized by down-regulation of mitotic spindle factors and upregulation of IL6-STAT3 signaling pathways.**

(A) Representative images of positive and negative correlations between protein abundances and rates of CIN of cB-ALL-PDX samples. (B) Histogram showing Spearman's rank correlations of protein abundances with CIN in cB-ALL-PDX samples. Dark-gray bars in the histogram depict the number of proteins significant correlating with CIN. (C) Significant GO terms for the positively (left) and negatively correlated (right) proteins. (D) UMAP of the different cB-ALL samples analyzed by RNAseq (EGAS00001003266). (E) Volcano plot representing the significantly enriched biological pathways up- (green) and downregulated (red) in aneuploid *versus* euploid cB-ALL samples. (F) Box-plots representing the gene set variation analysis (GSVA) of the indicated biological pathways downregulated (left) and upregulated (right) in the indicated cB-ALL subtypes. The box begins in the first quartile (percentile 25%) and ends in the third quartile (percentile 75%), central horizontal line represents the median value. Lines represent segments of furthest data without accounting for outliers. Two-way ANOVA tests were used. *$P < 0.05$, **$P < 0.01$; ***$P < 0.001$. Source data are available online for this figure.

observed microscopically in our cB-ALL-PDX models. In addition, the possibility of tracking karyotypes after each virtual cell division allowed us to delineate the dynamic karyotype evolution in the leukemic cell population over time. Although high levels of CIN can also slightly increase fitness and proliferation rates, it is important to underscore that high levels of CIN eventually lead to exacerbated chromosomal gains or losses, including a high frequency of monosomies that is not observed in reality, remarking again that low-intermediate CIN levels are optimal. Our results revealed a transient increase in karyotype heterogeneity, which is dramatically reduced over time, yielding more stable karyotypes that mainly exhibit intraclonal variability. This is consistent with the moderate levels of karyotype variability observed by scWGS (Bolhaqueiro et al, 2019; Nelson et al, 2020; Woodward et al, 2023).

A crucial determinant for CIN propagation is the capacity of cancer cells to tolerate a given level of genetic instability. In this sense, defects in different cellular pathways have been described to allow cancer cells to adapt to CIN. These defects include those that directly impinge on the chromosome segregation machinery, such as altered microtubule spindle dynamics, mechanisms required to correct erroneous kinetochore-microtubule attachments, centrosome clustering, and defects affecting the mitotic checkpoint (Bakhoum et al, 2009; Cohen-Sharir et al, 2021; Guo et al, 2022; Orr et al, 2016; Sansregret et al, 2017). The correlation of protein abundance with the rates of mitotic defects in our cB-ALL-PDX samples enabled us to interrogate the cellular mechanisms associated with CIN in cB-ALL. We found a proteomic signature characterized by the upregulation of several mitosis regulatory pathways, including those involved in regulation of microtubule cytoskeleton and chromosome segregation. Notably, we identified different proteins that were previously associated with the adaptation to CIN in cancer, including the kinesin-like protein KIF2C/MCAK and the master SAC regulator MPS1/TTK. Upregulation of KIF2C in CIN-positive cancer cells was found to increase microtubule turnover to counteract the hyperstable kinetochore-microtubule attachments that lead to lagging chromosomes (Bakhoum et al, 2009; Ertych et al, 2014; Orr et al, 2016). In addition, increasing mitosis duration by disrupting the SAC was also proposed as a cellular adaptation to CIN in cancer cells, most likely by allowing more time to correct improper kinetochore-microtubule attachments (Sansregret et al, 2017). Indeed, mitotic analyses identified mainly chromosome alignment and mitotic spindle defects in aneuploid cB-ALL-PDX samples, and both phenotypes are directly associated with kinetochore-microtubule attachment defects. Although it is possible that the upregulation of these genes reflects the higher mitotic activity observed in CIN-high samples and not CIN *per se*, we hypothesized that the upregulation of mitotic spindle regulators may reflect a cellular adaptation to

CIN in aneuploid cB-ALL cells and, hence, we would expect that aneuploid cB-ALL show a dysfunctional mitotic spindle. Accordingly, GO analyses of RNA-Seq data from patient cB-ALL samples revealed a significant down-regulation of the mitosis spindle signature in aneuploid samples. In addition, we found IL6-JAK-STAT3 signaling as the top upregulated pathway in aneuploid c-ALL. Notably, a recent study demonstrated that CIN relies on the IL6-STAT3 axis to prevent cell death, and that chemical blockade of IL-6 signaling impairs the survival of CIN+ breast cancer cells (Hong et al, 2022). The identification of drugs interfering with cell adaptation to CIN may open new avenues to explore CIN-targeted therapy in aneuploid cB-ALL, particularly in those subtypes with poor outcome. Potential drugs to be tested may include inhibitors of mitotic spindle factors, such as KIF2C/MCAK (GTSE1) or CENP-E (GSK-923295) (Serrano-Del Valle et al, 2021), inhibitors of centrosome clustering (AZ) (Guo et al, 2022), as well as inhibitors of the IL6-JAK-STAT signaling pathway (Tocilizumab, Tofacitinib and C188-9) (Johnson et al, 2018).

Altogether, our results reveal variable levels of ongoing CIN across aneuploid subtypes of cB-ALL, including HeH, HoL and NH, which are directly associated with karyotype heterogeneity and leukemia progression in preclinical PDX models. Our work highlights the risk of relying solely on the genome variability or on the expression of a small number of tumor markers to assess ongoing CIN in clinically relevant samples. Therefore, efforts should be made to identify reliable markers to assess ongoing CIN in routine clinical settings for risk-stratification and follow-up. In addition, our results identify a specific cellular signature associated with CIN in cB-ALL that may underlie an adaptation to overcome various cellular stresses imposed by it. This signature represents key vulnerabilities of leukemic cells with CIN. Modulating these stress pathways may provide new opportunities to target aneuploid cB-ALL with CIN.

## Methods

### Pediatric B-ALL leukemic samples

Diagnostic BM samples from pediatric patients with B-ALL ($n = 12$) were obtained from collaborating hospitals and from the VIVO biobank (UK). B-ALL diagnosis was based on French-American-British (FAB) and World Health Organization (WHO) classifications (Pui et al, 2004). Main cytogenetic/molecular diagnostics and other clinic-biological features of the patients are summarized in Table 1. All cB-ALL patiens included in this study did not contain classical subtype-defining fusions, including ETV6-RUNX1, TCF3-PBX1, KMT2A rearrangement, and BCL-ABL1. Samples were also negative for other oncogenic fusion

genes including ABL-class functions, JAK-class fusions, and CRLF2 rearrangements. Experiments conformed to the principles set out by the World Medical Association (WMA) Declaration of Helsinki and the Department of Health and Human Services Belmont Report. The study was approved by the Barcelona Clínic Hospital Institutional Review Ethics Board (HCB/2020/1347), and patient samples were accessed upon informed consent.

## Low-pass single-cell WGSeq

Single cells from primary cB-ALL samples were isolated using an inverted microscope coupled to a micromanipulator equipped with glass capillary for cell collection (Olympus IX71). A minimum of 20 cells were isolated per sample in microdrops of 2.5 μl of phosphate-buffered saline (PBS) with 0.5% polyvinyl alcohol (PVA). Cell lysis and DNA amplification were performed using the SurePlex DNA Amplification System (Illumina). Genomic DNA was subsequently fragmented and tagged with the VeriSeq PGS and Nextera XT index adapters by PCR for library preparation (VeriSeq PGS Library Prep Kit, Illumina). Equal volumes of normalized libraries were pooled and sequenced on an Illumina MiSeq platform with $1 \times 75$-bp single-end sequencing (target sequencing). Reads were subsequently aligned to the human reference genome (GRCh38/hg38) using Bowtie2 (version 2.2.4). The bam files, containing the aligned read data, were analyzed with the copy number calling algorithm AneuFinder (https://github.com/ataudt/aneufinder) (Bakker et al, 2016). Following GC correction and blacklisting of artefact-prone regions, libraries were analyzed using the *dncopy* copy number calling algorithm with variable width bind (average bin size = 1 Mb; step size = 500 kb). Libraries with on average less than 10 reads per bin were discarded. Whole-chromosome aneusomies were identified when >95% of the bins showed a deviation from the disomic state.

## PDX generation and follow-up

Seven- to 12-week-old nonobese diabetic/NOD.Cg-$Prkdc^{scid}$ $Il2rgt^{m1Wjl}$/$SzJ$ (NSG) mice (The Jackson Laboratory; RRID: IMSR_JAX:005557) ($n = 56$ with equal distribution of males and females), housed under pathogen-free conditions, were used in this study. All experimental procedures were approved by the Animal Care Committee of the Barcelona Biomedical Research Park (DAAM11883). A total of $5 \times 10^5$ primary blasts were used for intra-BM transplantation into sublethally irradiated (2 Gy) mice, as described (Molina et al, 2020). Leukemic engraftment was monitored in PB every other week from week 4 onwards by flow-cytometry using the monoclonal antibodies (mAbs) HLA-ABC fluorescein isothiocyanate (FITC; Clone G46-2.6; 1:100 dilution), CD19-phycoerythrin (PE; Clone HIB19; 1:100 dilution) and CD45-allophycocyanin (APC; Clone HI30; 1:100 dilution) (BD Biosciences). Mice were euthanized when leukemic engraftment reached 10 to 15%, typically representing >80% engraftment in BM (Molina et al, 2020), when disease symptoms were evident or at week 24. Blasts were isolated from BM and spleen by density-gradient centrifugation for downstream analyses. A minimum of three PDX (biological replicates) were generated for each primary

B-ALL sample. For EFS curves, an event was called when the human graft in PB was ≥0.1%. For OS curves, an event was considered when the leukemic graft was incompatible with animal welfare (>15% blasts in PB) in the absence of disease symptoms, or when disease symptoms were evident. The rates of proliferating and apoptotic cB-ALL blasts were analyzed at the end-point for each PDX by flow-cytometry using the mAb CD19-APC (Clone HIB19; 1:100 dilution) together with Ki67-FITC (clone B56; 2:100 dilution) or annexin V-PE (#556421; 5:100 dilution) (BD Biosciences), respectively.

## Immunofluorescence and microscopy analyses

Inmunofluorescence was performed as described on freshly-isolated cB-ALL cells from PDXs (Molina et al, 2020). In brief, $2 \times 10^6$ B-ALL cells were spun on poly-L-lysine-coated coverslips, fixed with 4% formaldehyde in PEM buffer (500 mM EGTA, 1 M $MgSO_4$, 500 mM PIPES pH 6.9, 2 M Sucrose) for 10 min at 37 °C and permeabilized with 0.2% Triton X-100-containing PEM buffer for 10 min at room temperature (RT). Cells were blocked with permeabilization buffer containing 1% bovine serum albumin (BSA) for 1 h at RT and incubated overnight at 4 °C with mouse anti-αtubulin (Sigma DM1A; 1:1000 dilution), rabbit anti-pericentrin (Abcam ab4448; 1:1000 dilution) and human anti-centromere (ACA; Antibodies Incorporated 15-234; 1:50 dilution) primary antibodies. Cells were washed with permeabilization buffer and incubated with fluorophore-conjugated anti-mouse Alexa555, anti-rabbit Alexa498 (ThermoFisher Scientific) and anti-human Cy5 (The Jackson Laboratory) secondary antibodies for 45 minutes at RT. Coverslips were mounted on slides with Vectashield 4',6-diamidino-2-phenylindol (DAPI; Vector Laboratories). Immuno-fluorescence analyses were performed using a Leica AF6000 motorized microscopy system (Leica Microsystems, Manheim, Germany) equipped with a Leica DMI6000 inverted microscope, a Leica PL APO 63× numerical aperture 1.4 oil immersion, a high-resolution monochroma Hamamatsu Orca ER C4742-80 Digital Camera and a mercury metal halide bulb Leica EL6000 as light source. Image acquisition was performed using the software LasX Navigator. DAPI was acquired with a band pass excitation filter 340/380 nm, dichromatic mirror (400 nm) and a long-pass emission filter (425 nm), Alexa488 was acquired with a band pass excitation filter 480/40 nm, dichromatic mirror 505 nm and a band pass emission filter (527/30 nm), Alexa555 was acquired with a band pass excitation filter 531/40 nm, dichromatic mirror reflection 499–555 and transmission 659–730 nm and a band pass emission filter (593/40 nm). Cy5 was acquired with excitation band pass filter 628/40 nm, dichromatic mirror reflection 594–651 nm and transmission 699–726 nm and a band pass emission filter (692/40 nm).

For mitotic index quantification, ten random fields were captured, and quantification of interphase and mitotic cells were performed in Z-stack projections using the Cell Counter plugin in Fiji-ImageJ (NIH). Between 1295 and 3694 cells were scored, and the percentage of mitotic cells was registered in every single experiment. For mitosis progression analyses, a minimum of 200 mitotic cells were analyzed per experiment to assess the frequency of mitotic cells in each mitotic phase and the frequency of mitotic defects.

## Multicolor fluorescence in situ hybridization on metaphase chromosomes

Freshly-isolated B-ALL cells were cultured for 16 h in Stemspan medium (Stem Cell Technologies, Vancouver, Canada) supplemented with 20% fetal calf serum, the hematopoietic cytokines SCF (100 ng/mL), FLT3 ligand (100 ng/mL), IL3 (10 ng/mL), and IL7 (10 ng/mL) (all from PeproTech), insulin-transferrin selenium and antibiotics (Gibco). Cells were subsequently treated with 0.2 µg/mL of the tubulin polymerization inhibitor Colcemid (Gibco) for 3 h before harvesting. Metaphase chromosome spreads were obtained following standard cytogenetic procedures (Molina et al, 2012). In brief, colcemid-treated cells were collected and resuspended in pre-warmed 75 mM KCl hypotonic solution for 10 min at 37 °C and subsequently fixed in freshly prepared methanol:acetic acid (3:1) solution. Preparations were processed for M-FISH using the 24XCyte Human Multicolor FISH Probe kit (MetaSystems Probes). M-FISH capturing was performed in a Metafer Slide Scanning System (MetaSystems) with a AX10 ZEISS epifluorescence microscope equipped with a motorized stage, 10× and 63× oil plan APOCHROMAT objectives and specific filters for DAPI, Spectrum Green, Spectrum Orange, Spectrum Aqua, Spectrum Red, Spectrum Far Red and Spectrum Gold (Nikon). M-FISH analyses were performed on the Isis FISH imaging system (MetaSystems). Karyotype Heterogeneity score (kHS) was calculated for each sample using the formula $kHS = \frac{\sum_{i=1}^{N} \frac{i\,\text{Observed copies}}{i\,\text{Expected copies}}}{N}$, where $N$ is the total number of homologous chromosomes ($N = 23$).

## In silico modeling of chromosome instability

### Fitness per chromosome determination

To build an unbiased fitness value for every given karyotype, we considered different features related to B-cell hematopoietic differentiation, B-ALL, tumor development and genome structural features that may influence the rates of specific chromosome mis-segregation. First, we included a total of 239 genes extracted from different GO pathways, including hematopoietic progenitor cell differentiation (GO:0002244), B-cell activation (GO:0042113), B-cell differentiation (GO:0030183), and lymphocyte activation (GO:0046649). To calculate the fitness value per chromosome associated with these genes, defined as B-ALL related fitness ($\alpha$), the chromosomal localization of the genes was identified and normalized by the maximum gene density value per chromosome. Second, to include genes with a demonstrated contribution to ALL pathogenesis, we also considered the density of ALL-associated somatic gene mutations per chromosome. A total of 117 genes from the COSMIC database records for "lymphoblastic leukemia" were used. We discarded 13 genes that were already present for the B-ALL-related fitness cost determination. These genes were annotated based on their chromosomal location to define the B-ALL somatic mutations fitness ($\beta$). Finally, we also considered the PANcancer approximation proposed by Davoli et al (2013) (Davoli et al, 2013) regarding the density of oncogenes (oGs), tumor suppressor genes (TSGs) and essential genes (eGs). Thus, we defined the total driver density fitness ($\gamma$) by computing the sum of these three density values, as previously done by others (Lynch et al, 2022).

In addition, evidence is mounting that the probability of specific chromosomes undergoing a segregation error is non-random

(Klaasen and Kops, 2022). Some structural chromosome features are known to affect chromosome mis-segregation, including chromosome size, centromere size and density of CENPB-box sequences, gene density or chromosome location in the interphase nucleus (Drpic et al, 2018; Fachinetti et al, 2015; Klaasen et al, 2022; Worrall et al, 2018). To take the non-random probability of chromosome mis-segregation, we included available data on the centromeric size ($S^{Cen}$) and total gene density ($S^{GD}$) per chromosome in our model (Jabalameli et al, 2019; Mayer et al, 2005) (Fig. 6B and Appendix Table S2). Taking this information into consideration, we built the following expression to calculate global cell fitness ($\Phi$, Fig. 6B): $\Phi = \sum_{i=1}^{23} \omega_1 \left( \frac{\alpha_i + \beta_i + \gamma_i}{3} \right) + \omega_2 \frac{S_i^{GD}}{S_i^{Cen}}$. Preferentially gained or retained chromosomes in aneuploid B-ALL were not included in our simulations to avoid overfitting and biases in our B-ALL CIN agent-based.

### Agent-based model for in silico analyses of chromosomal instability

The agent-based model was implemented in MATLAB (R2022a, MathWorks, Inc., Natick, MA, USA). Data analysis was performed in R (R 4.2.2, R Foundation for Statistical Computing, Vienna, Austria) and MATLAB. Due to the computational cost of tracking complete karyotypes at those time steps for each individual cell, the carrying capacity of the system was set to $10^6$ cells. Total simulation time was adjusted to 100 days. Results were stored every 6 iterations (12 h) and 50 independent simulations were conducted for each $P_{CIN}$ value. An initial standard population of 500 aneuploid cells was generated following a normal distribution $\mathcal{N}(47, 1)$. For patient-based simulations, we took the modal karyotypes obtained by single-cell next-generation sequencing in primary samples as a seed to generate the initial population. Cell division was modeled stochastically based on a $P_{DIV}$, which was randomly picked in the interval [0.2, 0.4] day$^{-1}$ (Clarkson et al, 1967; Lynch et al, 2022) following a normal distribution. During cell division, chromosome mis-segregation could occur with a fixed probability $P_{CIN}$ in the interval $P_{DIV}$ [0, 0.4] day$^{-1}$. Preferentially gained or retained chromosomes in aneuploid B-ALL were not included in our simulations at this point to avoid overfitting and biases in our B-ALL CIN agent-based. Please note that since chromosome instability is linked to cell proliferation, we are considering the product $P_{div} \cdot P_{CIN}$. Cells having less than 1 copy or more than 6 would immediately undergo mitotic catastrophe and would be removed from the system. Dividing cells showing a chromosome count below 40 chromosomes may undergo whole-genome duplication (WGD). Although WGD has been extensively reported in advanced cancer patients, affecting around 30% of them (Bielski et al, 2018), the experimental evidence reporting the WGD rate at a single-cell resolution is scarce. We decided to set the probability of successfully undergoing WGD to $P_{WGD} = 0.011$ day$^{-1}$. $P_{DIV}$ is updated at each time step based on the average fitness value of the previous time step according to $P_{DIV}(t) = P_{DIV}(t-1)(1 + \Delta\overline{\Phi})$. In addition, every cell may undergo cell death naturally with a fixed probability $P_{DEATH}$ (Appendix Table S3). All these probabilities were multiplied by a factor of $\frac{1}{\Delta t}$ and turned into rates at each time step, which was set to 2 h due to computational cost. Cell fitness ($\Phi$) was calculated for each individual cell in each simulation time step, by multiplying its karyotype by the contribution of each chromosome copy to cell fitness (Appendix Table S3). We worked under the assumption that unbalanced chromosome stoichiometry

positively contributes to cell adaptability (Chen et al, 2015). To prevent a selection bias in favor of high-hyperdiploid cells, we internally normalized cell fitness by the total number of chromosomes in each time step. Parameter values and the conceptual frame for our model are summarized in Fig. 6A and Appendix Table S3.

## Mass spectrometry and proteomic analyses

Whole-cell lysates (WCL) of $2 \times 10^6$ freshly-isolated cB-ALL cells from PDX samples were obtained using a lysis buffer containing 9 M Urea, 20 mM HEPES pH 8.0, 1 mM sodium orthovanadate, 2.5 mM sodium pyrophosphate and 1 mM β-glycerophosphate. Protein samples (10 μg) were reduced with dithiothreitol (30 nmol, 37 °C, 60 min) and alkylated in the dark with iodoacetamide (60 nmol, 25 °C, 30 min). The resulting protein extract was first diluted to 2 M urea with 200 mM ammonium bicarbonate for digestion with endoproteinase LysC (1:10 w-w, 37 °C, 6 hours; Wako), and then diluted 2-fold with 200 mM ammonium bicarbonate for trypsin digestion (1:10 w-w, 37 °C, overnight; Promega). After digestion, the peptide mix was acidified with formic acid and desalted with a MicroSpin C18 column (The Nest Group, Inc.) prior to LC-MS/MS analysis.

Samples were analyzed using a Orbitrap Eclipse mass spectrometer (Thermo Fisher Scientific, San Jose, CA, USA) coupled to an EASY-nLC 1200 (Thermo Fisher Scientific (Proxeon), Odense, Denmark). Peptides were loaded directly onto the analytical column and were separated by reversed-phase chromatography using a 50-cm column with an inner diameter of 75 μm, packed with 2 μm C18 particles. Chromatographic gradients started at 95% buffer A and 5% buffer B with a flow rate of 300 nl/min and gradually increased to 25% buffer B and 75% A in 79 min and then to 40% buffer B and 60% A in 11 minutes. After each analysis, the column was washed for 10 minutes with 100% buffer B. Buffer A: 0.1% formic acid in water. Buffer B: 0.1% formic acid in 80% acetonitrile.

The mass spectrometer was operated in positive ionization mode with nanospray voltage set at 2.4 kV and source temperature at 305 °C. The acquisition was performed in data-dependent acquisition mode and full MS scans with 1 micro scans at resolution of 120,000 were used over a mass range of $m/z$ 350–1400 with detection in the Orbitrap mass analyzer. Auto gain control (AGC) was set to 'standard' and injection time to 'auto'. In each cycle of data-dependent acquisition analysis, following each survey scan, the most intense ions above a threshold ion count of 10,000 were selected for fragmentation. The number of selected precursor ions for fragmentation was determined by the "Top Speed" acquisition algorithm and a dynamic exclusion of 60 seconds. Fragment ion spectra were produced via high-energy collision dissociation at normalized collision energy of 28%, and they were acquired in the ion trap mass analyzer. AGC was set to $2 \times 10^4$ and an isolation window of 0.7 $m/z$ and a maximum injection time of 12 ms were used.

Digested BSA (New England Biolabs) was analyzed between each sample to avoid sample carryover and to assure stability of the instrument, and qCloud was used to control instrument longitudinal performance during the project (Chiva et al, 2018). Acquired spectra were analyzed using Proteome Discoverer software suite (v2.0, ThermoFisher Scientific) and the Mascot search engine v2.6 (Matrix Science) (Perkins et al, 1999). The data were searched against a Swiss-Prot human database (https://www.uniprot.org, April 2022, 20,401 entries) plus a list of common contaminants and all the corresponding decoy entries (Beer et al, 2017). For peptide identification, a precursor ion mass tolerance of 7 ppm was used for MS1 level, trypsin was chosen as enzyme, and up to three missed cleavages were allowed. The fragment ion mass tolerance was set to 0.5 Da for MS2 spectra. Oxidation of methionine and N-terminal protein acetylation were used as variable modifications, whereas carbamidomethylation on cysteines was set as a fixed modification. False discovery rate (FDR) in peptide identification was set to a maximum of 1%. Peptide quantification data were retrieved from the "Precursor ion area detector" node from Proteome Discoverer (v2.5) using 2 ppm mass tolerance for the peptide extracted ion current. Protein abundances were calculated as the average of the three most abundant distinct peptide groups and normalized based on total peptide amount. For the group comparison analysis, protein normalized abundances from Proteome Discoverer were Log2-transformed prior to calculation of fold change, $P$-value, and adjusted $P$-value ($q$-value).

The normalized values of protein abundance were compared against the heterogeneity score of each sample and a correlation coefficient was calculated using Pearson correlation test, using the *cor.test* function in R. Proteins with a Pearson's correlation coefficient >0.5 or <−0.5 and with a $P$-value < 0.01 were grouped as positively or negatively correlated, respectively. Within each group (that is positively and negatively correlated proteins), a dendrogram was calculated using the heatmap.2 function in R. An Euclidean distance was computed both between proteins and samples and clustered using default hierarchical clustering. Finally, the enrichment of GO terms within each group was carried out using the *TOPGO* method implemented in R (Alexa et al, 2006) using the "*weight01*" algorithm. GO terms with an adjusted $P$-value < 0.05 were considered as significant and plotted using ggplot.

## RNA-sequencing

To analyze the gene expression profiling from patients, we selected the dataset of St. Judes Hospital from the European Genome-phenome Archive (EGA) under accession number EGAS00001003266. This dataset contains data from 1988 patients with B-ALL. Prior to the analysis, we re-classified each patient based on the number of chromosomes and the copy-number alterations by RNAseq. Aneuploids (high-hyperdiploid, low-hyperdiploid, low-hypodiploid, and near-haploid) and euploid ($n = 46$ chromosomes with variable genomic reorganizations) B-ALL patients were selected for comparison. Data from adult patients were also filtered out. A total of 765 patients were analyzed after re-classification. The log2 (FPKM) expression data of these patients were downloaded.

Data pre-processing, exploration, and differential gene expression (DGE) analyses were performed with DESeq2 (v1.38.1) R package. We used principal component analysis and UMAP dimensionality reduction to perform an explorative data analysis. Using the sva package, we could remove batch effects caused by the library preparation and sequencing lengths. DGE analysis was performed with the Wald test (p.adj Benjamini–Hochberg correction <0.05). A total of 699 significant DEGs were found in the comparison between aneuploid and euploid samples; of these, 115

**The paper explained**

**Problem**

Chromosomal instability (CIN) is a prominent form of genomic instability and a major cause of aneuploidy, a hallmark of cancer. Aneuploidy is commonly associated with ongoing CIN through consecutive cell divisions, resulting in intratumoral chromosome copy-number heterogeneity (chr-CNH), a central driver of cancer evolution and ultimately, unfavorable clinical outcomes. Despite its ubiquitous presence in cancer, the presence of CIN in B-cell acute lymphoblastic leukemia (B-ALL) remains unknown due to the lack of preclinical models to study actively diving cells. B-ALL is the most common cancer in children (cB-ALL) and aneuploidy is the most common genetic alteration, with specific subtypes conferring important prognostic values.

**Results**

We elucidated the presence of CIN in aneuploid cB-ALL subtypes using single-cell whole-genome sequencing of primary cB-ALL samples and by generating and functionally characterizing patient-derived xenograft models (cB-ALL-PDX). Our results show higher rates of CIN across aneuploid cB-ALL than in euploid cB-ALL that strongly correlate with intraclonal chr-CNH and overall survival in mice. This association was further supported by in silico mathematical modeling. In addition, proteomic analyses of cB-ALL-PDX revealed a CIN "signature" enriched in mitotic-spindle regulatory pathways, which was confirmed by RNA-sequencing of a large cohort of cB-ALL samples.

**Impact**

The link between the presence of CIN in aneuploid cB-ALL and disease progression opens new possibilities for patient stratification and offers a promising new avenue as a therapeutic target in cB-ALL treatment with aneuploidies. Thus, CIN levels could serve as a potent biomarker for patient classification, empowering clinicians to make informed decisions for more personalized and effective treatment strategies.

had log2 FC > 0 (overexpressed in the aneuploid samples) the remaining 584 genes with log2 FC < 0. DGE results were further explored by Over Representation Analysis and Gene Set Enrichment Analysis (GSEA). We queried different databases including GO and Molecular Signatures Database (mSigDB). Both analyses were performed using the clusterProfiler (v 4.4.4) package.

## Statistical analyses

Statistical comparisons were performed using R-statistics v4.0.0 (R Foundation for Statistical Computing, Vienna, Austria) or GraphPad Prism version 6.0 (GraphPad Software). All data were analyzed according to the test indicated in the appropriate figure legends on the indicated number of experiments. Kolmogorov–Smirnov's test was used to assume normal distributions prior to statistical comparisons by either parametric or non-parametric tests. The levels of significance were as follows: $*P < 0.05$, $**P < 0.01$, and $***P < 0.001$.

Mice were randomized on the day of irradiation to establish identical or similar group sizes with a minimum of 3 mice transplanted with each primary cB-ALL sample. No statistical methods were used to predetermine sample sizes, but they were based on our previous publications (Lopez-Millan et al, 2019; Molina et al, 2020; Prieto et al, 2018). Animal technicians were blinded to sample identity. No further blinding was performed.

Mice were censored from analyses in rare instances when sacrificed for non-leukemia reasons.

## Data availability

Single-cell Whole-Genome Sequencing data of cB-ALL patients: EGAD50000000029 (https://ega-archive.org/). Modeling computer scripts: GitHub (https://github.com/molabEvoDynamics/OscarMolinaEtAl_CINandB-ALL). Mass spectrometry proteomic data: PRIDE PXD042785.

## Peer review information

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

## Acknowledgements

The authors thank Anthony V Moorman (University of Newcastle, UK) for assistance in recruiting primary samples, Maria Calvo and Gemma Martín (Scientific and Technological Centers, Universitat de Barcelona [CCiTUB]), Angelika Merkel (Bioinformatics Unit, Institut de Recerca Josep Carreras) for computational technical assistance, Eva Borràs and Eduard Sabidó (Proteomics Facility, Universitat Pompeu Fabra/Centre de Regulació Genòmica de Barcelona) for technical assistance on mass spectrometry and proteomic analyses. We thank CERCA program (Generalitat de Catalunya) and the Josep Carreras Foundation-Obra Social La Caixa for core support. Financial support for this work was obtained from the Spanish Ministry of Economy and Competitiveness/European Union NextGenerationEU (PID2022-142966OB-I00) to PM and OM, the Deutsche José Carreras Leukämie-Stiftung (DJCLS 15R/2023) to OM and ISCIII-RICORS-TERAV within the Next Generation EU program (plan de recuperación, transformación y resiliencia) (RICORS, RD21/0017/0029) to PM. Additional funding was provided by the ISCIII (FEDER PI17/01028 and PI20/00822) to CB and the Generalitat de Catalunya (2022/SGR-003) to PM. OM was supported by the Asociación Española contra el Cancer (AECC; INVES211226MOLI). COS and TVH were supported by the AECC (2019-PRED-28372 and INVES223069VELA, respectively). The work of COS, GFC and VMP-G was supported by the Spanish Ministerio de Ciencia e Innovación and the European Union NextGenerationEU/PRTR, MCIN/AEI/10.13039/501100011033 (grant numbers PID2019-110895RB-I00, PID2022-142341OB-I00, TED2021-132296B-C55, PDC2022-133520-I00). PM is an investigator of the Spanish Cell Therapy Cooperative Network (TERCEL).

## Author contributions

**Oscar Molina**: Conceptualization; Data curation; Formal analysis; Supervision; Funding acquisition; Validation; Investigation; Visualization; Methodology; Writing—original draft; Project administration; Writing—review and editing. **Carmen Ortega-Sabater**: Data curation; Formal analysis; Investigation; Methodology. **Namitha Thampi**: Investigation. **Narcís Fernández-Fuentes**: Data curation; Investigation; Methodology. **Mercedes Guerrero-Murillo**: Investigation. **Alba Martínez-Moreno**: Investigation. **Meritxell Vinyoles**: Investigation. **Talía Velasco-Hernández**: Investigation. **Clara Bueno**: Investigation; Methodology. **Juan L Trincado**: Investigation. **Isabel Granada**: Investigation; Methodology. **Diana Campos**: Resources. **Carles Giménez**: Resources. **Judith M Boer**: Resources. **Monique L den Boer**: Resources. **Gabriel F Calvo**: Supervision; Investigation; Methodology. **Mireia Camós**: Resources. **Jose-Luis Fuster**: Resources. **Pablo Velasco**: Resources. **Paola Ballerini**: Resources. **Franco Locatelli**: Resources. **Charles G Mullighan**: Resources. **Diana CJ Spierings**: Resources; Methodology. **Floris Foijer**: Resources; Methodology. **Victor M Pérez-García**: Supervision; Investigation; Methodology. **Pablo Menéndez**: Supervision; Funding acquisition; Validation; Writing—review and editing.

## Disclosure and competing interests statement

The authors declare no competing interests.

# Expanded View Figures

**Figure EV1.  Aneuploid cB-ALL show higher rates of mitotic defects and chromosomal clonal heterogeneity in PDX models.**

(**A**) Box-plots representing the percentage of mitotic defects in the indicated B-ALL PDX samples ($n = 36$ PDX, $n = 3$ PDX per primary sample; $n = 200$ mitotic cells per PDX sample, total $= 7200$ mitotic cells). The box begins in the first quartile (percentile 25%) and ends in the third quartile (percentile 75%), central horizontal line represents the median value. Vertical line represents segment of furthest data from minimum (bottom) to maximum (top) values. (**B**) Frequency of mitotic PDX-expanded primary blasts from (**A**) with the indicated mitotic defects ($n = 36$, 3 cB-ALL-PDX samples per leukemia; $n = 200$ mitotic cells per PDX sample, Total$=7200$ mitotic cells). (**C**) Box-plots representing the percentage of late mitosis defects in the indicated cB-ALL PDX samples ($n = 36$ PDX samples; $n = 123$ Eup, 116 HeH, 163 HoL and 186 NH late mitoses). The box begins in the first quartile (percentile 25%) and ends in the third quartile (percentile 75%), central horizontal line represents the median value. Vertical line represents segment of furthest data from minimum (bottom) to maximum (top) values. (**D**) Chromosome number distributions of cells in the indicated B-ALL PDX samples as determined by M-FISH on metaphase spreads. Center values indicate the median and error bars indicate the SEM. Number of cells analyzed and chromosome modal numbers (MN) are indicated at the top. Dashed line shows the normal euploid chromosome number. (**E**) Chromosome number distributions of cells in matching primary and PDX samples, as determined by scWGS (blue) and M-FISH (red). Center values indicate the median and error bars indicate the SEM. (**F**) Chromosome MN in the indicated B-ALL PDX samples as determined by M-FISH analyses from (**D**). Red dashed lines show the disomic (2n) copy-number. (**G**) Box-plots representing the variability of chromosome copy-numbers observed by M-FISH analyses from (**D**), as determined by the standard deviation (SD) in the indicated ploidy group. The box begins in the first quartile (percentile 25%) and ends in the third quartile (percentile 75%), central horizontal line represents the median value. Vertical line represents segment of furthest data from minimum (bottom) to maximum (top) values.

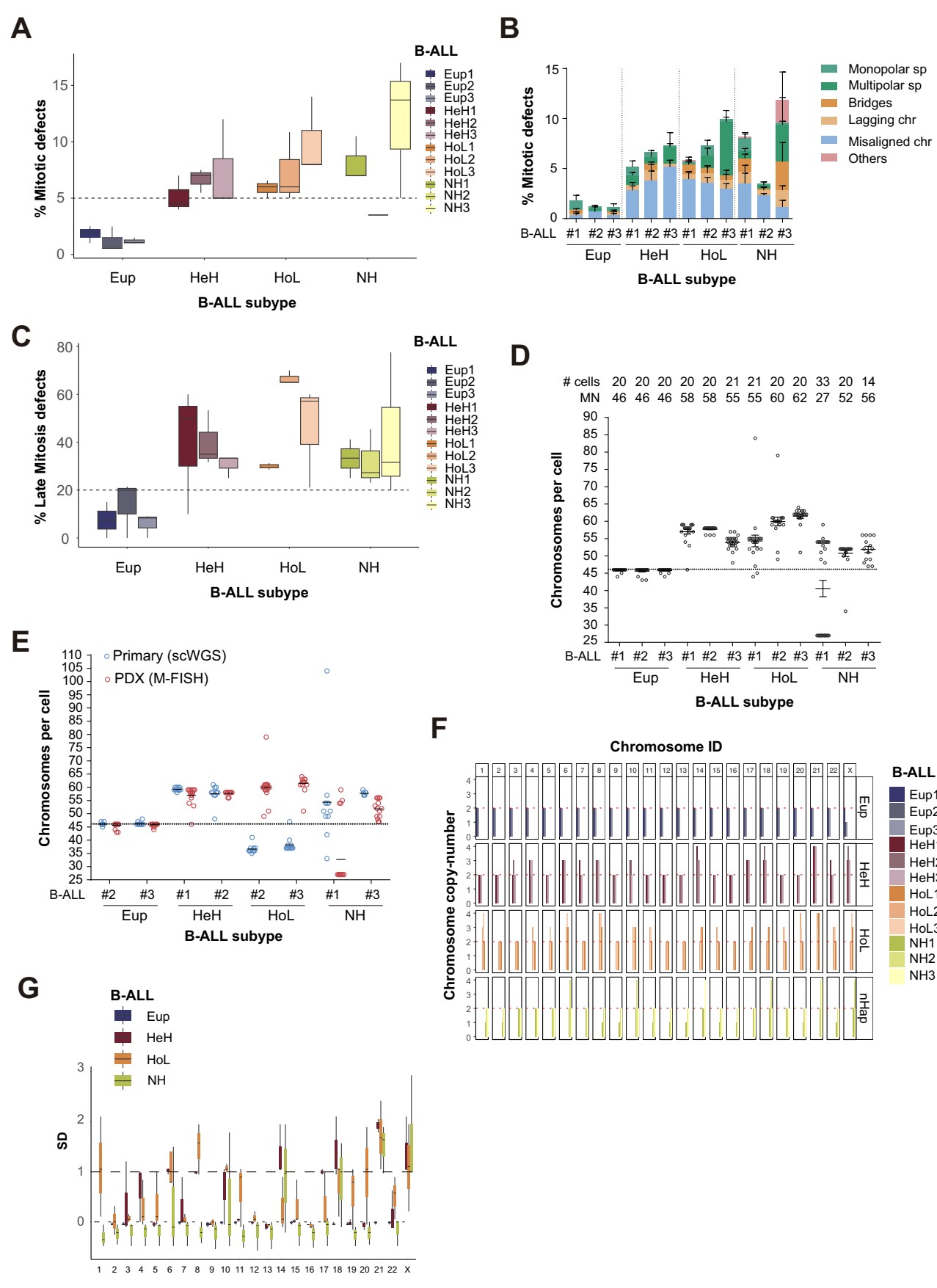

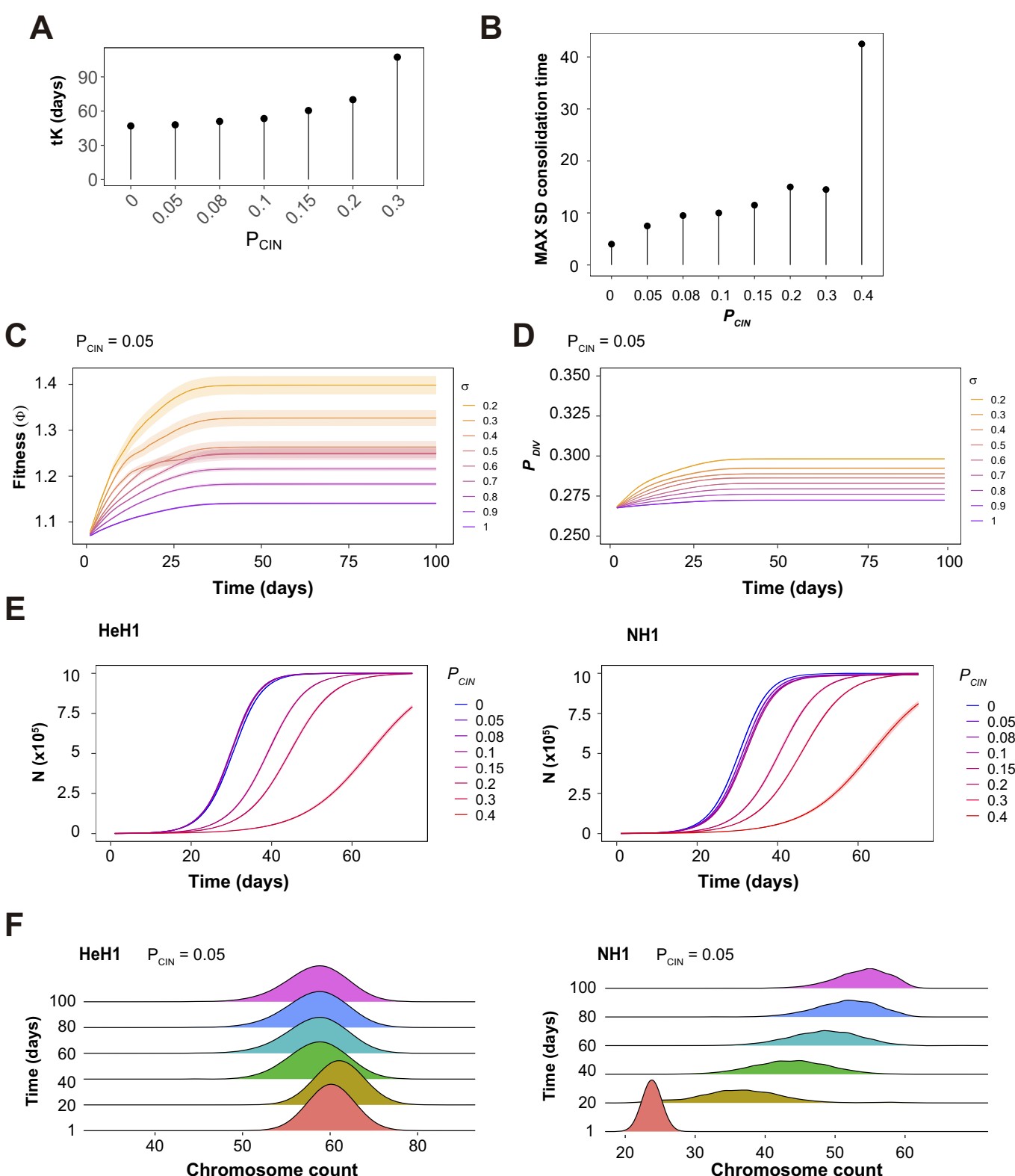

**Figure EV2. Mathematical modeling associates low-to-mid levels of CIN as drivers of clonal heterogeneity and disease progression in cB-ALL.**

(A) Average time to reach carrying capacity (tK) at the indicated CIN levels ($N = 50$ simulations). (B) Time to consolidate a stable karyotype at the indicated CIN levels ($N = 50$ simulations). (C) Simulated dynamics of average fitness at the indicated CIN levels ($N = 50$ simulations). (D) Simulated dynamics of cell division rates at the indicated CIN levels ($N = 50$ simulations). (E) Simulated cell numbers for virtual sample HeH1 (left) and NH1 (right) at the indicated CIN levels ($N = 20$ simulations). Average fitness is plotted relative to that expected for a euploid cell, the latter being equal to 1. (F) Karyotype variability as observed by chromosome counts at the indicated time points from in silico simulations with the virtual HeH1 (left) and NH1 (right) samples at moderate CIN levels ($P_{CIN} = 0.05$).

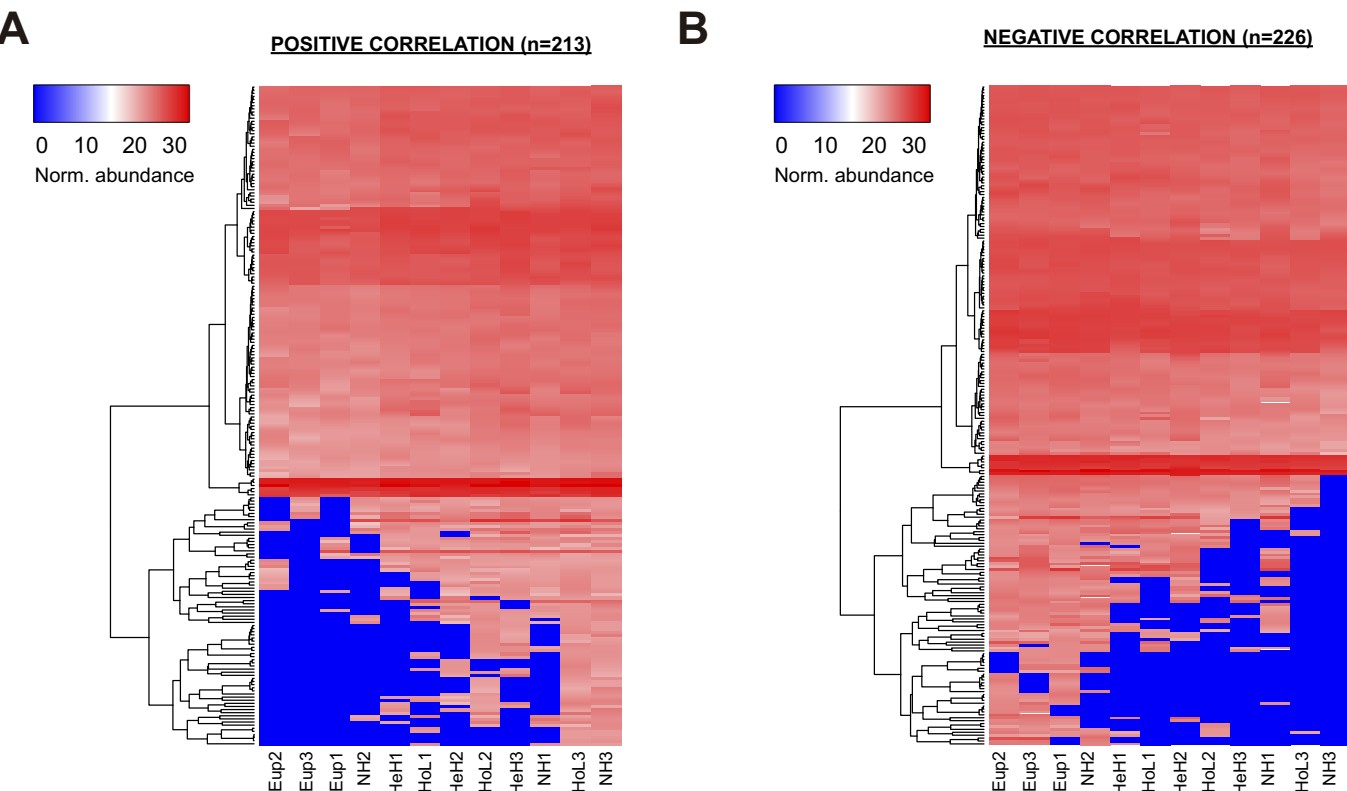

**Figure EV3. Proteins associated with CIN in cB-ALL.**

(A, B) Heatmaps depicting proteins positively (A) and negatively (B) correlated with CIN. Protein abundances in whole-cell lysates are color-represented in the legends (top). Pearson correlation coefficient ($P < 0.05$).

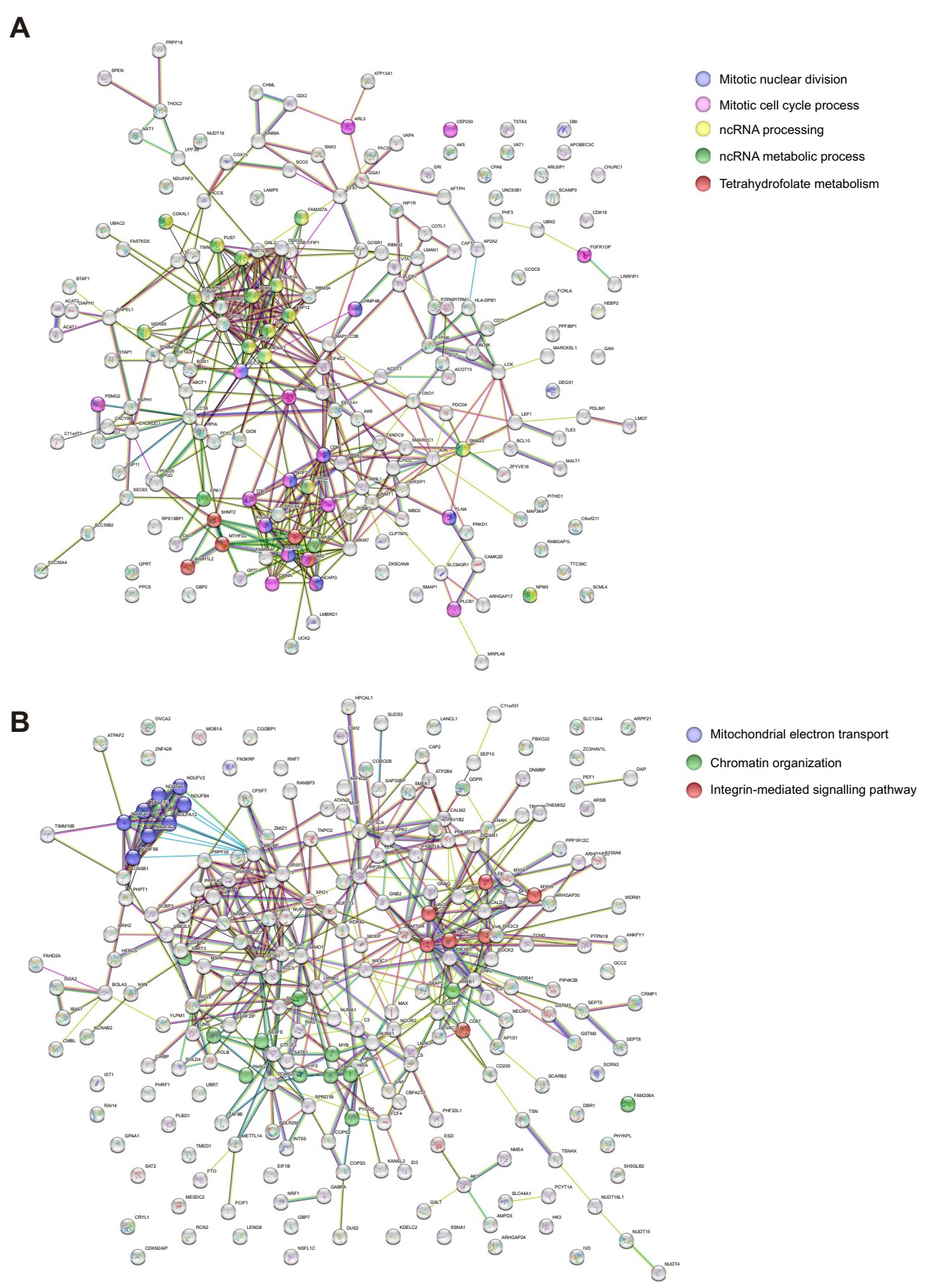

**Figure EV4. Protein cluster analyses associated with CIN in cB-ALL.**

(A, B) Protein–protein interaction network analysis with proteins positively (**A**) and negatively (**B**) correlated with CIN using the STRING database (version 11.5). Protein clusters are colored according with the indicated GO pathways. Protein–protein interaction (PPI) enrichment $P$-value $= 1 \times 10^{-16}$ (**A**) and $1.67 \times 10^{-15}$ (**B**).

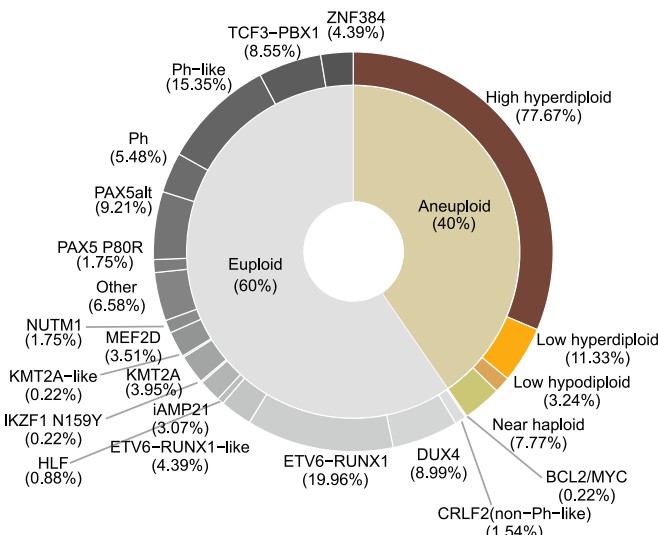

**Figure EV5.   Genetic subtypes of patients analyzed by RNA sequencing.**

Pie chart depicting the frequency of individual cB-ALL molecular subgroups identified in the RNA-Seq St Judes hospital cohort of cB-ALL samples.

