## [Peer Review File · EMBO Molecular Medicine]

CHROMOSOME INSTABILITY IN ANEUPLOID ACUTE LYMPHOBLASTIC LEUKEMIA ASSOCIATES WITH DISEASE PROGRESSION

Oscar Molina, Carmen Ortega-Sabater, Namitha Thampi, Narcis Fernandez-Fuentes, Mercedes Guerrero-Murillo, Alba Martínez, Meritxell Vinyoles, Talia Velasco-Hernandez, Clara Bueno, Juan Trincado, Isabel Granada, Diana Campos, Carles Giménez, Judith Boer, Monique denBoer, Gabriel Calvo, Mireia Camós, Jose Fuster, Pablo Velasco, Paola Ballerini, Franco Locatelli, Charles Mullighan, Diana Spierings, Floris Foijer, Victor Perez-Garcia, and Pablo Menendez

DOI: [10.15252/emmm.202318468](https://doi.org/10.15252/emmm.202318468)

Corresponding authors: Oscar Molina (omolina@carrerasresearch.org) , Pablo Menendez (pmenendez@carrerasresearch.org)

Review Timeline:

Submission Date:	5th Aug 23
Editorial Decision:	31st Aug 23
Revision Received:	19th Oct 23
Editorial Decision:	31st Oct 23
Revision Received:	9th Nov 23
Accepted:	15th Nov 23

Editor: Lise Roth

Transaction Report:

31st Aug 2023

Dear Dr. Molina,

Thank you for the submission of your manuscript to EMBO Molecular Medicine. We have now received feedback from the three reviewers who agreed to evaluate your manuscript. As you will see from the reports below, the referees acknowledge the interest of the study and are overall supporting publication of your work pending appropriate minor revisions.

Addressing the reviewers' concerns in full will be necessary for further considering the manuscript in our journal, and acceptance of the manuscript will entail a second round of review. EMBO Molecular Medicine encourages a single round of revision only and therefore, acceptance or rejection of the manuscript will depend on the completeness of your responses included in the next, final version of the manuscript. For this reason, and to save you from any frustrations in the end, I would strongly advise against returning an incomplete revision.

We are expecting your revised manuscript within three months, if you anticipate any delay, please contact us.

We require:

4) A .docx formatted letter INCLUDING the reviewers' reports and your detailed point-by-point responses to their comments. As part of the EMBO Press transparent editorial process, the point-by-point response is part of the Review Process File (RPF), which will be published alongside your paper.

5) A complete author checklist, which you can download from our author guidelines (<https://www.embopress.org/page/journal/17574684/authorguide#submissionofrevisions>). Please insert information in the checklist that is also reflected in the manuscript. The completed author checklist will also be part of the RPF.

6) Please note that all corresponding authors are required to supply an ORCID ID for their name upon submission of a revised manuscript. An ORCID identifier is currently missing for Prof. Pablo Menendez.

7) It is mandatory to include a 'Data Availability' section after the Materials and Methods. Before submitting your revision, primary datasets produced in this study need to be deposited in an appropriate public database, and the accession numbers and database listed under 'Data Availability'. Please remember to provide a reviewer password if the datasets are not yet public (see <https://www.embopress.org/page/journal/17574684/authorguide#dataavailability>).

8) For data quantification: please specify the name of the statistical test used to generate error bars and P values, the number (n) of independent experiments (specify technical or biological replicates) underlying each data point and the test used to calculate p-values in each figure legend. The figure legends should contain a basic description of n, P and the test applied. Graphs must include a description of the bars and the error bars (s.d., s.e.m.). Please provide exact p values.

9) Our journal encourages inclusion of *data citations in the reference list* to directly cite datasets that were re-used and obtained from public databases. Data citations in the article text are distinct from normal bibliographical citations and should directly link to the database records from which the data can be accessed. In the main text, data citations are formatted as

follows: "Data ref: Smith et al, 2001" or "Data ref: NCBI Sequence Read Archive PRJNA342805, 2017". In the Reference list, data citations must be labeled with "[DATASET]". A data reference must provide the database name, accession number/identifiers and a resolvable link to the landing page from which the data can be accessed at the end of the reference. Further instructions are available at .

13) Author contributions: CRediT has replaced the traditional author contributions section because it offers a systematic machine readable author contributions format that allows for more effective research assessment. Please remove the Authors Contributions from the manuscript and use the free text boxes beneath each contributing author's name in our system to add specific details on the author's contribution. More information is available in our guide to authors.

14) Every published paper now includes a 'Synopsis' to further enhance discoverability. Synopses are displayed on the journal webpage and are freely accessible to all readers. They include a short stand first (maximum of 300 characters, including space) as well as 2-5 one-sentences bullet points that summarizes the paper. Please write the bullet points to summarize the key NEW findings. They should be designed to be complementary to the abstract - i.e. not repeat the same text. We encourage inclusion of key acronyms and quantitative information (maximum of 30 words / bullet point). Please use the passive voice. Please attach these in a separate file or send them by email, we will incorporate them accordingly.

15) As part of the EMBO Publications transparent editorial process initiative (see our Editorial at <http://embomolmed.embopress.org/content/2/9/329>), EMBO Molecular Medicine will publish online a Review Process File (RPF) to accompany accepted manuscripts.

In the event of acceptance, this file will be published in conjunction with your paper and will include the anonymous referee reports, your point-by-point response and all pertinent correspondence relating to the manuscript. Let us know whether you agree with the publication of the RPF and as here, if you want to remove or not any figures from it prior to publication.

I look forward to receiving your revised manuscript.

Yours sincerely,

Lise Roth

***** Reviewer's comments *****

Referee #1 (Comments on Novelty/Model System for Author):

This is a solid manuscript that represents a sizable amount of work. The link between CIN and disease progression is not strictly novel, but this provides an in-depth analysis of this relationship in a pediatric cancer and will be of interest to cancer researchers.

Referee #1 (Remarks for Author):

Overall, this is an interesting study that provides an in-depth dive into chromosomal instability in pediatric leukemia. The single-cell sequencing analysis is a strength and furthers our understanding of this disease. I believe that this manuscript represents a basically-complete piece of work, and I would be happy to recommend acceptance of the manuscript without any additional experimental work. I would suggest the following minor changes for the authors to consider:

I don't agree with the following statement: "phase I clinical studies assessing the efficiency of inhibitors of MPS1, a master regulator of the SAC, and of KIF18A, a kinesin-like motor protein that regulates chromosome positioning during cell division, are currently being conducted to treat aneuploid cancers with ongoing CIN (NCT02366949, NCT04293094)." - so far as I'm aware, these are Phase I clinical trials testing these compounds across a variety of cancer types, and are not using CIN as a biomarker for trial admission. Academic research has suggested that these compounds may be most effective in high-CIN cancers, but I don't believe that that has influenced the design of these trials. So saying that these trials are "being conducted to treat aneuploid cancers with ongoing CIN" is inaccurate.

Can the authors clarify: do they know the TP53 status of these leukemias? If not, what frequency of p53 mutations have been reported in the literature for this cancer type? P53 and CIN are tightly linked, can they comment on any connection here?

I think that Figure 1 should include a table that defines each leukemia subtype. The discussion of the subtype definitions in the next could also be clarified.

To me, it was quite striking how consistent certain chromosome gain events were in the single-cell sequencing data, despite the presence of CIN. It seemed like there was more variability among chromosome loss events than chromosome gain events. Would you expect gains and losses to appear at similar frequencies? Could this be related to the "addiction" idea that cancer cells would die if they lost these chromosome gains (PMID: 37410869)?

Figure 2C/D - the colors used in the KM plots were difficult for me to distinguish.

I would suggest carefully changing some of the language and discussion regarding the upregulation of mitotic genes as detected via mass spec and RNA-Seq. The expression of these chromosome segregation genes correlates with mitotic activity (see, for instance, PMID: 22028643 and 28320919). And, the high-CIN cancers have higher mitotic activity than the low-CIN cancers (Fig. 3A). So, it's possible that the upregulation of these genes reflects the higher mitotic activity and not CIN per se. It is tempting to call this a "CIN signature", but the analysis here does not allow one to determine whether the upregulation of these genes causes CIN or is a consequence of higher mitotic activity (and is actually driven by something else, like a shared mutation). I think that this circularity should be directly addressed.

Referee #2 (Comments on Novelty/Model System for Author):

High impact data which will significantly advance the field. Results are novel and will have a strong translational impact by advancing both molecular diagnosis and targeted therapy for cB-ALL. The experiments were well executed, using a cutting-edge technology and analysis.

Referee #2 (Remarks for Author):

In the submitted manuscript, the authors study the role of chromosomal instability (CIN) in progression of childhood B-cell acute lymphoblastic leukemia (cB-ALL). The authors determined the presence of CIN in aneuploid cB-ALL subtypes using single-cell whole-genome sequencing of primary cB-ALL samples and by generating and functionally characterizing patient-derived xenograft models (cB-ALL-PDX). They identified higher rates of CIN across aneuploid than in euploid cB-ALL, which strongly correlated with intraclonal chromosomal copy number heterogeneity (chr-CNH) and overall survival in mice. The mass-spectrometry analyses of cB-ALL-PDX, along with RNA-sequencing revealed a CIN signature enriched in mitotic-spindle regulatory pathways. Based on these data, the authors conclude that the link between the presence of CIN in aneuploid cB-ALL and disease progression can lead to the novel patient stratification and the identification of the new therapeutic targets for cB-ALL treatment.

Elucidating molecular mechanisms that regulate progression of childhood B-cell acute lymphoblastic leukemia (cB-ALL) is an important topic. Results are novel and will have a strong translational impact by advancing both molecular diagnosis and targeted therapy for cB-ALL. Thus, these results are of high importance to the broad readership of the EMBO Molecular Medicine journal. The experiments were well executed, using a cutting-edge technology and analysis. The authors are the leaders in the field. This reviewer has a few minor suggestions: 1) The authors should indicate if any of the patients with cB-ALL was diagnosed as Ph-like ALL; 2) The authors should provide a list of the genes listed in mitotic spindle and IL6 JAK-STAT signaling groups in Figure 7 E-F (as Supplemental table). This will allow easier access to the important data for a broad audience of EMBO Molecular Medicine; and 3) Discussion should be expanded to include potential drugs which could be tested in cB-ALL with CIN based on the presented data (e.g. inhibitors of JAK-STAT pathway or drugs that target mitotic spindle). Thus, due to the high novelty and translational significance, this manuscript has a high potential to significantly advance the field. The overall novelty and significance of the results overcome the minor concerns, that were outlined above. Addressing the minor concern would make the manuscript more complete and suitable for a broad readership of the EMBO Molecular Medicine.

Referee #3 (Remarks for Author):

The gain or loss of whole chromosomes, termed aneuploidy, was identified as a distinct feature of cancer cells more than a century ago by the German zoologist Theodor Boveri, and is now recognized as a major genomic insult in human cancers. Aneuploidy is observed more frequently than any other oncogenic or tumor-suppressor mutation, and is found in ~90% of solid tumors and ~60% of hematological malignancies. Aneuploidy is common in paediatric B-cell precursor acute lymphoblastic leukaemia (ALL). Specific subgroups, such as high hyperdiploidy (>50 chromosomes or DNA Index {greater than or equal to}1.16) and hypodiploidy (<45 chromosomes), predict outcome of patients after primary treatment. Under the 2022 WHOHAEM 5 classification, precursor B-cell neoplasms are classified based on ploidy changes, such as hyperdiploidy and hypodiploidy, as well as chromosomal rearrangements or the presence of other genetic drivers. In this study, Authors explain the presence of CIN in aneuploid cB-ALL subtypes using single-cell whole-genome sequencing of primary cB-ALL samples and by generating and functionally characterizing patient-derived xenograft models (cB-ALL-PDX). They reported higher rates of CIN across aneuploid than in euploid cB-ALL that strongly correlate with intraclonal chr-CNH and overall survival in mice. What is more, they used mass-spectrometry to analyze of cB-ALL-PDX and revealed a CIN signature enriched in mitotic-spindle regulatory pathways, which was confirmed by RNA-sequencing of a large cohort of cB-ALL samples.

The format of the work is properly constructed and the content of the study includes important observations and conclusions.

The used methods are appropriate and they were described very well in the manuscript. I have two suggestions

1. I have one suggestion. If you had data from SNP array, we would can observe the presence of CIN in molecular karyotypes of patients.
2. Table 2 in supplementary - please correct notation of karyotype Hol#2 M-FISH for PDX model

REFEREE #1

Referee #1 (Comments on Novelty/Model System for Author):

This is a solid manuscript that represents a sizable amount of work. The link between CIN and disease progression is not strictly novel, but this provides an in-depth analysis of this relationship in a pediatric cancer and will be of interest to cancer researchers.

We thank the referee for his/her compliments.

Referee #1 (Remarks for Author):

Overall, this is an interesting study that provides an in-depth dive into chromosomal instability in pediatric leukemia. The single-cell sequencing analysis is a strength and furthers our understanding of this disease. I believe that this manuscript represents a basically-complete piece of work, and I would be happy to recommend acceptance of the manuscript without any additional experimental work.

We thank the referee for his/her compliments.

I would suggest the following minor changes for the authors to consider:

I don't agree with the following statement: "phase I clinical studies assessing the efficiency of inhibitors of MPS1, a master regulator of the SAC, and of KIF18A, a kinesin-like motor protein that regulates chromosome positioning during cell division, are currently being conducted to treat aneuploid cancers with ongoing CIN (NCT02366949, NCT04293094)." - so far as I'm aware, these are Phase I clinical trials testing these compounds across a variety of cancer types, and are not using CIN as a biomarker for trial admission. Academic research has suggested that these compounds may be most effective in high-CIN cancers, but I don't believe that that has influenced the design of these trials. So saying that these trials are "being conducted to treat aneuploid cancers with ongoing CIN" is inaccurate.

We agree with the referee that this statement in the introduction was not completely accurate. We have now rephrased this statement as "phase I clinical studies assessing the efficiency of inhibitors of MPS1, a master regulator of the SAC, and of KIF18A, a kinesin-like motor protein that regulates chromosome positioning during cell division, are currently being conducted to treat a variety of cancer types (NCT02366949, NCT04293094)." (Page 3, line 91).

Can the authors clarify: do they know the TP53 status of these leukemias? If not, what frequency of p53 mutations have been reported in the literature for this cancer type? P53 and CIN are tightly linked, can they comment on any connection here?

We thank the referee for his/her academic input. We agree with the referee that *TP53* mutations are tightly linked to CIN in most cancer types. However, genetic alterations affecting *TP53* are uncommon in cB-ALL at diagnosis, with their incidence being lower than 5% (PMID: 31894096; 31729120). An exception is the rare low hypodiploid (HoL) B-ALL subtype (31-39 chromosomes), with approximately 90% of the patients' harbouring mutations or deletions affecting *TP53* (PMID: 35008193). Indeed, as expected our three HoL have *TP53* mutations. For this reason, cB-ALL samples are not tested for *TP53* mutations at diagnosis in a routine clinical setting, except in rare cases with a HoL or suspected endoreduplicated HoL karyotype. Therefore, *TP53* mutations are unlikely to have a direct contribution to CIN in most cB-ALL subtypes. Notwithstanding, due to the strong association of *TP53* mutations with CIN in other cancers, we agree with the referee that it should be commented in the MS. We have now included a comment in the introduction (**page 4; lines 104-106**) and further clarified it in the results section (**page 14; line 386**) in the revised version of our MS.

I think that Figure 1 should include a table that defines each leukemia subtype. The discussion of the subtype definitions in the next could also be clarified.

Following the referee's suggestion, we have now included a definition of each leukemia subtype in the new version for **Figures 1B** and **2A**. To further clarify this point, we included a definition for each leukemia subtype in the legend of Table 1 (**page 39**) in the revised version of our MS.

To me, it was quite striking how consistent certain chromosome gain events were in the single-cell sequencing data, despite the presence of CIN. It seemed like there was more variability among chromosome loss events than chromosome gain events. Would you expect gains and losses to appear at similar frequencies? Could this be related to the "addiction" idea that cancer cells would die if they lost these chromosome gains (PMID: 37410869)?

We agree with the referee that certain chromosome gains in the single-cell WGS data (as well as in the M-FISH analyses) are very consistent. Indeed, as we state throughout the manuscript, CIN rates translate into moderate levels of intraclonal chr-CNH in aneuploid cB-ALL samples. This situation has been reported in other cancer types and organoid models, where low-to-mid karyotypic changes have been reported despite the presence of widespread CIN (PMID: 31036964; PMID: 32054838), indicating that cell intrinsic and extrinsic factors play a crucial role in the selection of specific karyotype changes. As the referee pointed out, patients with chromosome loss events seem to be more variable than others with chromosome gain events. Although we cannot rule out the "addiction" idea of some cancer cells to specific trisomies, we believe that this situation better respond to i) the sample HoL2, which is one of these samples, is the one with the lowest percentage of leukemic cells (30%). As expected, we could detect a population with chromosome loss events (blasts) and another population with euploid karyotypes (healthy blood cells, which were not considered for calculating the

heterogeneity score) in our single-cell WGS data; or ii) both low hypodiploid and near haploid subtypes show chromosome endoreduplication events very often (PMID: 35008193), as we could observe in some of our samples despite the relatively low number of cells analysed (i.e. NH1 patient, second cell (row) is near haploid and the rest have exactly or almost exact doubled-up chromosome content).

Figure 2C/D - the colors used in the KM plots were difficult for me to distinguish.

Following the referee's suggestion, we have now modified the KM plots in **Figure 2C** and **D** so colours in these plots can now be easily distinguished.

I would suggest carefully changing some of the language and discussion regarding the upregulation of mitotic genes as detected via mass spec and RNA-Seq. The expression of these chromosome segregation genes correlates with mitotic activity (see, for instance, PMID: 22028643 and 28320919). And, the high-CIN cancers have higher mitotic activity than the low-CIN cancers (Fig. 3A). So, it's possible that the upregulation of these genes reflects the higher mitotic activity and not CIN per se. It is tempting to call this a "CIN signature", but the analysis here does not allow one to determine whether the upregulation of these genes causes CIN or is a consequence of higher mitotic activity (and is actually driven by something else, like a shared mutation). I think that this circularity should be directly addressed.

We thank the referee for raising this academic point. We have now clarified this important point in the discussion section: "Although it is possible that the upregulation of these genes reflects the higher mitotic activity observed in CIN-high samples and not CIN per se, we hypothesized..." (**Page 19, lines 489-490**). In addition, we included " " when talking about the CIN signature (abstract **Page 2, line 61**; introduction, **Page 5, line 134**) to avoid overstatements.

REFEREE #2

Referee #2 (Comments on Novelty/Model System for Author):

High impact data which will significantly advance the field. Results are novel and will have a strong translational impact by advancing both molecular diagnosis and targeted therapy for cB-ALL. The experiments were well executed, using a cutting-edge technology and analysis.

We thank the referee for his/her compliments.

Referee #2 (Remarks for Author):

In the submitted manuscript, the authors study the role of chromosomal instability (CIN) in progression of childhood B-cell acute lymphoblastic leukemia (cB-ALL). The authors determined the presence of CIN in aneuploid cB-ALL subtypes using single-cell whole-genome sequencing of primary cB-ALL samples and by generating and functionally characterizing patient-derived xenograft models (cB-ALL-PDX). They identified higher rates of CIN across aneuploid than in euploid cB-ALL, which strongly correlated with intraclonal chromosomal copy number heterogeneity (chr-CNH) and overall survival in mice. The mass-spectrometry analyses of cB-ALL-PDX, along with RNA-sequencing revealed a CIN signature enriched in mitotic-spindle regulatory pathways. Based on these data, the authors conclude that the link between the presence of CIN in aneuploid cB-ALL and disease progression can lead to the novel patient stratification and the identification of the new therapeutic targets for cB-ALL treatment.

Elucidating molecular mechanisms that regulate progression of childhood B-cell acute lymphoblastic leukemia (cB-ALL) is an important topic. Results are novel and will have a strong translational impact by advancing both molecular diagnosis and targeted therapy for cB-ALL. Thus, these results are of high importance to the broad readership of the EMBO Molecular Medicine journal. The experiments were well executed, using a cutting-edge technology and analysis. The authors are the leaders in the field.

We thank the referee for his/her compliments.

This reviewer has a few minor suggestions:

- 1) The authors should indicate if any of the patients with cB-ALL was diagnosed as Ph-like ALL.

Ph-like is a specific subgroup of ALL characterized by a similar transcriptomic profile as Ph-positive patients in the absence of the BCR-ABL fusion gene (PMID: 19138562). Ph-like profile is mostly associated with “B-other” B-ALL lacking hallmark genetic alterations found in B-ALL such as BCR-ABL1, ETV6-RUNX1, TCF3-PBX1, KMT2A (MLL) rearrangements, high hyperdiploidy or hypodiploidy, etc (PMID: 19138562). Standard diagnostic procedures to classify a patient as Ph-like include the detection of ABL-class fusion, JAK-class fusion, CRLF2 rearrangements or IKZF1 deletions (PMID: 25207766, PMID: 19138562). No a single patient in our cohort contained classical B-ALL subtype-defining fusions as assessed by standard diagnostic procedures. Samples were also negative for other oncogenic fusion genes defining most Ph-like cases including ABL-class and JAK-class fusions and CRLF2 rearrangements. Therefore, none of the patients were diagnosed as Ph-like ALL.

To clarify this point, we have now rephrased the Material and Methods to read as follows: “All cB-ALL patients included in this study lacked classical subtype-defining fusions, including ETV6-RUNX1, TCF3-PBX1, KMT2A rearrangements and BCR-ABL1. Samples were also negative for other oncogenic fusion genes including ABL-class and JAK-class fusions and CRLF2 rearrangements” (**Page 21, lines 521-524**).

2) The authors should provide a list of the genes listed in mitotic spindle and IL6 JAK-STAT signalling groups in Figure 7 E-F (as Supplemental table). This will allow easier access to the important data for a broad audience of EMBO Molecular Medicine.

Following the referee’s suggestion, we have now included a table (**Appendix Table S5**) containing the genes listed in the mitotic spindle and IL6-JAK-STAT signalling GO pathways. We included the p-value of these genes after comparing Euploid vs Aneuploid patients as done in Figure 7 E-F.

3) Discussion should be expanded to include potential drugs which could be tested in cB-ALL with CIN based on the presented data (e.g. inhibitors of JAK-STAT pathway or drugs that target mitotic spindle).

Following the referee’s suggestion, we have now included a brief expansion on the discussion providing specific examples of potential drugs that can be tested in cB-ALL with high-CIN (**Page 19, lines 498-501**): “Potential drugs to be tested may include inhibitors of mitotic spindle factors, such as KIF2C/MCAK (GTSE1) or CENP-E (GSK-923295) (PMID: 34129859), inhibitors of centrosome clustering (AZ) (PMID: 36612150), as well as inhibitors of the IL6-JAK-STAT signalling pathway (Tocilizumab, Tofacitinib and C188-9) (PMID: 29405201)”.

Thus, due to the high novelty and translational significance, this manuscript has a high potential to significantly advance the field. The overall novelty and significance of the results overcome the minor concerns, that were outlined above. Addressing the minor concern would make the manuscript more complete and suitable for a broad readership of the EMBO Molecular Medicine.

We thank the referee for his/her academic review.

REFEREE #3

Referee #3 (Remarks for Author):

The gain or loss of whole chromosomes, termed aneuploidy, was identified as a distinct feature of cancer cells more than a century ago by the German zoologist Theodor Boveri, and is now recognized as a major genomic insult in human cancers. Aneuploidy is observed more frequently than any other oncogenic or tumor-suppressor mutation, and is found in ~90% of solid tumors and ~60% of hematological malignancies. Aneuploidy is common in paediatric B-cell precursor acute lymphoblastic leukaemia (ALL). Specific subgroups, such as high hyperdiploidy (>50 chromosomes or DNA Index {greater than or equal to}1.16) and hypodiploidy (<45 chromosomes), predict outcome of patients after primary treatment. Under the 2022 WHOHAEM 5 classification, precursor B-cell neoplasms are classified based on ploidy changes, such as hyperdiploidy and hypodiploidy, as well as chromosomal rearrangements or the presence of other genetic drivers. In this study, Authors explain the presence of CIN in aneuploid cB-ALL subtypes using single-cell whole-genome sequencing of primary cB-ALL samples and by generating and functionally characterizing patient-derived xenograft models (cB-ALL-PDX). They reported higher rates of CIN across aneuploid than in euploid cB-ALL that strongly correlate with intraclonal chr-CNH and overall survival in mice. What is more, they used mass-spectrometry to analyze of cB-ALL-PDX and revealed a CIN signature enriched in mitotic-spindle regulatory pathways, which was confirmed by RNA-sequencing of a large cohort of cB-ALL samples.

The format of the work is properly constructed and the content of the study includes important observations and conclusions. The used methods are appropriate and they were described very well in the manuscript.

We thank the referee for his/her compliments.

I have two suggestions:

1. I have one suggestion. If you had data from SNP array, we would can observe the presence of CIN in molecular karyotypes of patients.

Indeed, copy-number variability by SNP array data was used in Figure 1 to assess karyotype variability in the indicated cB-ALL samples (Appendix Table S1), showing a moderate significant increase in chr-CNH in aneuploid versus euploid karyotypes (Figure 1B).

2. Table 2 in supplementary - please correct notation of karyotype Hol#2 M-FISH for PDX model.

We corrected the karyotype formula for HoL#2 M-FISH for PDX model in **Appendix Table S2**. We apologise for the typo and acknowledge the referee's ability to detect it.

31st Oct 2023

Dear Dr. Molina,

Thank you for submitting your revised manuscript and point-by-point rebuttal to the referees' minor comments. I am pleased to say that I will be able to accept your manuscript once the following editorial points will be addressed:

1/ Manuscript text:

- Please remove the red text and only keep in track changes mode any new modification.
- We noted discrepancies in the following authors' name: Narcís Fernández in the manuscript file vs. Narcis Fernandez-Fuentes in the submission system; Alba Martínez-Moreno in the manuscript file vs. Alba Martínez in the submission system. Please clarify.
- Please rename "Methods" to "Materials and methods"
 - o Patients' samples: please include the full sentence that the experiments conformed to the principles set out in the WMA Declaration of Helsinki and the Department of Health and Human Services Belmont Report.
 - o Animals: please provide information on gender and age of the mice.
 - o Antibodies: please make sure to indicate the used dilutions/concentrations for all antibodies.
- Data Availability section: Please provide an accession number and URL link for all datasets. Note that the data must be publicly available at the time of publication.
- Acknowledgements: Please make sure that the information provided in the manuscript matches the information provided in the submission system (ISCI-RICORS-TERAV within the Next Generation EU program (plan de recuperación, transformación y resiliencia); missing in eJP: (FEDER PI17/01028 and PI20/00822), Asociación Española contra el Cáncer (AECC; INVES211226MOLI), AECC (2019-PRED-28372 and INVES223069VELA), Spanish Ministerio de Ciencia e Innovación and the European Union NextGenerationEU/PRTR, MCIN/AEI/10.13039/501100011033 (grant numbers PID2019-110895RB-I00, PID2022-142341OB-I00, TED2021-132296B-C55, PDC2022-133520-I00). PM is an investigator of the Spanish Cell Therapy cooperative network (TERCEL) are currently missing from the submission system).
- Please rename "Conflict of interest" to "Disclosure Statement and Competing Interests".

2/ Figures and Appendix:

- Please provide exact p values, not a range, in the figures or in their legends, including for ns - non-significant.
- The separately uploaded Appendix Table S1 and S5 need to be renamed and uploaded as Dataset EV1 and EV2 - the callouts in the manuscript need editing too
- The Appendix should be a PDF file.
- In the figure legends:
 - Please note that legend for figure EV2f is missing; however, the corresponding figure panel is provided. This needs to be rectified.
 - Please indicate the statistical test used for data analysis in the legends of figures EV3a-b; EV4a-b.
 - Please note that the box plots need to be defined in terms of minima, maxima, centre, bounds of box and whiskers, and percentile in the legend of figures 1b; 5d; 7f; V1a, c, g
 - Please note that information related to n is missing in the legend of figures EV1b
 - Please indicate what yellow arrows represent in legend of figure 3c

3/ I slightly edited your synopsis text to fit our format, please let me know if you agree with the following:

Chromosomal instability (CIN) is a form of genomic instability and a major cause of aneuploidy, which associates with unfavorable clinical outcomes. The presence of CIN remains unknown in pediatric B-cell acute lymphoblastic leukemia (cB-ALL), where aneuploidy is the most common genomic abnormality.

- The different aneuploid subtypes of cB-ALL show moderately higher levels of chromosomal copy-number heterogeneity than cB-ALL with euploid karyotypes.
- Primary-derived xenograft (PDX) models of cB-ALL consistently show higher levels of mitotic defects that correlate with chromosomal copy-number heterogeneity and lower overall survival rates.
- Intracлонаl karyotype variability in one or few dominant clones expanded during leukemogenesis shape the chromosomal copy-number heterogeneity spectrum in aneuploid cB-ALL.
- Mathematical modelling of CIN in cB-ALL confirms a relationship between CIN and leukemia progression.
- Cellular adaptation to CIN involves the mitotic spindle machinery to regulate chromosome segregation and to maintain the levels of CIN under a tolerable threshold.

4/ As part of the EMBO Publications transparent editorial process initiative (see our Editorial at <http://embomolmed.embopress.org/content/2/9/329>), EMBO Molecular Medicine will publish online a Review Process File (RPF) to accompany accepted manuscripts.

This file will be published in conjunction with your paper and will include the anonymous referee reports, your point-by-point

response and all pertinent correspondence relating to the manuscript. Let us know whether you agree with the publication of the RPF.

I look forward to receiving your revised manuscript.

Yours sincerely,

Lise Roth

The authors addressed the minor editorial issues.

15th Nov 2023

Dear Dr. Molina,

Thank you for providing the revised files. I am pleased to inform you that your manuscript is accepted for publication and is now being sent to our publisher to be included in the next available issue of EMBO Molecular Medicine!

Congratulations on your interesting work!

Yours sincerely,

Lise Roth
